# Can't See the Wood for the Trees: Can Visual Adversarial Patches Fool Hard-Label Large Vision-Language Models?

## Abstract

Large vision-language models (LVLMs) have demonstrated impressive capabilities in handling multi-modal downstream tasks, gaining increasing popularity. However, recent studies show that LVLMs are susceptible to both intentional and inadvertent attacks. Existing attackers ideally optimize adversarial perturbations with backpropagated gradients from LVLMs, thus limiting their scalability in practical scenarios as real-world LVLM applications will not provide any LVLM's gradient or details. Motivated by this research gap and counter-practical phenomenon, we propose the first and novel hard-label attack method for LVLMs, named *HardPatch*, to generate visual adversarial patches by solely querying the model. Our method provides deeper insights into how to investigate the vulnerability of LVLMs in local visual regions and generate corresponding adversarial substitution under the practical yet challenging hard-label setting. Specifically, we first split each image into uniform patches and mask each of them to individually assess their sensitivity to the LVLM model. Then, according to the descending order of sensitive scores, we iteratively select the most vulnerable patch to initialize noise and estimate gradients with further additive random noises for optimization. In this manner, multiple patches are perturbed until the altered image satisfies the adversarial condition. Extensive LVLM models and datasets are evaluated to demonstrate the adversarial nature of the proposed *HardPatch*. Our empirical observations suggest that with appropriate patch substitution and optimization, *HardPatch* can craft effective adversarial images to attack hard-label LVLMs.

## 1 Introduction

Nowadays, large vision-language models (LVLMs) (Bai et al., 2023; Ye et al., 2023), at the juncture of computer vision and natural language processing, have become indispensable and marked a significant milestone in the field of artificial intelligence. By further benefiting from the strong comprehension of large language models (LLMs) (Brown et al., 2020; Touvron et al., 2023a;b), recent LVLMs (Dai et al., 2024; Liu et al., 2024a; Zhu et al., 2023) on top of LLMs show notable developments in numerous downstream tasks (Nichol et al., 2021; Ramesh et al., 2022; Rombach et al., 2022; Tsimpoukelli et al., 2021; Li et al., 2023; Alayrac et al., 2022). However, most recently proposed LVLMs suffer from severe security issues (Liu et al., 2024b; Fan et al., 2024), where an attacker's well-crafted adversarial input sample can easily fool the LVLM models, posing a considerable challenge to real-world LVLM applications.

Based on the accessibility level of victim models, existing LVLM attackers can be generally categorized into three types: white-box attacks (Bailey et al., 2023; Dong et al., 2023; Fu et al., 2023; Cui et al., 2023; Gao et al., 2024a; Wang et al., 2024; Lu et al., 2024; Luo et al., 2024; Gao et al., 2024b), gray-box attacks (Shayegani et al., 2023; Wang et al., 2023), and transfer-based black-box attacks (Zhao et al., 2024; Yin et al., 2023; Guo et al., 2024), as shown in Figure 1 (a). For white-box attacks, the attackers are assumed to have full knowledge of the victim LVLMs, including model architecture and parameters. These works simply formulate the attack as an optimization problem and utilize the backpropagated gradient to generate adversarial examples. To alleviate this reliance on model details to a certain extent, gray-box attacks solely require access to the visual encoder of LVLMs. However, since real-world LVLM applications are impossible to share any model details

with users, white-/gray-box attacks seem excessively idealistic and cannot work well in practical scenarios. Although no target-model details are required in transfer-based black-box attacks, they still rely on the additional knowledge of other surrogate LVLM models. In sum, existing LVLM attackers are severely limited by their scalability, and there is no attack that truly does not require any prior LVLM information in a more challenging hard-label setting (Cheng et al., 2018).

To address this research gap, we introduce the first hard-label adversarial attack against LVLMs, where the attackers can solely query the input/output of LVLMs. However, without using model details, it is difficult to determine where and how to add perturbations to images to mislead LVLMs. Luckily, the design of adversarial patch provides a concise and interpretable way to achieve successful real-world attacks (Brown et al., 2017; Duan et al., 2020). By appropriately placing the adversarial patches on the image according to the model's attention, its adversarial nature will fool the LVLM's eyes and lead to inaccurate prompt reasoning. Moreover, we empirically find that adversarial patches have fewer perturbations and are easier to add than directly perturbing pixel-wise noises on whole images (Zhao et al., 2024; Cheng et al., 2018), as shown in Figure 1 (b). Based on the above observations, we attempt to investigate *"How to design effective adversarial patches to mislead hard-label LVLMs?"*. Therefore, the remaining questions in designing LVLM attacks are: In the hard-label setting, (1) how to explore the LVLM's attention on different local regions of images for patch substitution? and (2) how to design/optimize the patch pattern in order to achieve the adversarial condition?

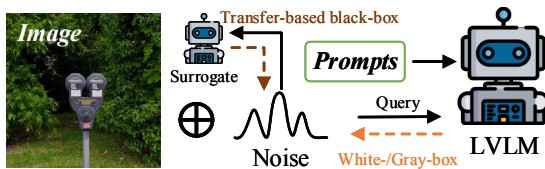

(a) Existing attackers need prior LVLM knowledge.

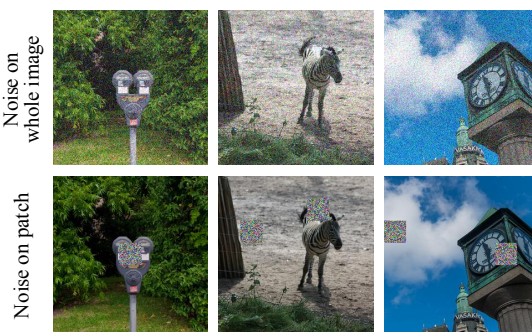

(b) Comparison: Global noise *vs*. Adv. Patches.

Figure 1: (a) Attack process of existing LVLM attackers. (b) We re-implement (Zhao et al., 2024) in the hard-label setting by removing its surrogate model. Compared to it, our adversarial patches have fewer perturbations and are easier to add.

In this paper, we propose a novel adversarial patch method called *HardPatch* to tackle the above hard-label issues. Specifically, we first uniformly split the input image into multiple patches with the same size. Then, to assess the sensitivity of each patch to the LVLM model, we individually mask each patch and feed them into the LVLM to measure the semantic changes between their corresponding text output and the original output. The larger the distance, the more sensitive the LVLM model is to altering the corresponding patch. Therefore, by scoring all patches according to their sensitivities in descending order, we iteratively substitute the more vulnerable patch with initial noise and estimate gradients with further additive random noises for optimizing the adversarial pattern. If the patch updated with a fixed number of iterations is still not adversarial, we additionally perform the same altering process on the next patch. Multiple patches are perturbed until the altered image satisfies the adversarial condition. The key contributions of our work are outlined as follows: (i) We design *HardPatch*, a novel adversarial attack method for more practical yet challenging hard-label LVLMs. We propose to generate visual adversarial patches to be added to input images for attackers in real-world scenarios. (ii) To determine where to place the adversarial patch, we develop a replacement order determination module to investigate the sensitivity of LVLM to each patch. Based on this, we iteratively substitute more vulnerable patches with noise and design the gradient estimation strategy to further optimize it until the attack succeeds. (iii) These insights are validated by extensive experiments on different LVLM models and datasets. Corresponding results demonstrates the effectiveness of our proposed *HardPatch* against hard-label LVLM models.

## 2 RELATED WORK

**Adversarial Robustness of LVLM Models.** LVLMs generally combine the capabilities of processing visual information with natural language understanding by using pre-trained vision encoders

with language models. Due to this multimodal nature (Szegedy et al., 2013), LVLMs are particularly vulnerable as the multi-modal integration not only amplifies their vulnerable utility but also introduces new attack vectors that are absent in unimodal systems. Most of existing LVLM attackers (Bailey et al., 2023; Dong et al., 2023; Fu et al., 2023; Cui et al., 2023; Gao et al., 2024a; Wang et al., 2024; Lu et al., 2024; Luo et al., 2024; Gao et al., 2024b) are inspired by the adversarial vulnerability observed in vision tasks. They evaluate the adversarial robustness of LVLMs under white-box settings, where they have the full knowledge of LVLMs models including network structure and weights. To generate the adversarial examples, they simply add and optimize imperceptible perturbations on the whole image to benign image inputs via back-propagation. To reduce the reliance on model knowledge, some gray-box attackers (Shayegani et al., 2023; Wang et al., 2023) solely require access to the visual encoder of LVLMs and directly generate the perturbed visual representations to fool the latter process. Although a few researchers (Zhao et al., 2024; Yin et al., 2023; Guo et al., 2024) claim that they achieve more challenging black-box attacks, their attacks are implemented in a transfer-based setting, where they still require the additional knowledge of other surrogate LVLM models to generate adversarial samples then transfer them to attack victim LVLMs. Therefore, how to design an LVLM adversarial attack in a more practical hard-label setting is still a research gap.

**Adversarial Patch.** Adversarial patches (Brown et al., 2017; Karmon et al., 2018; Eykholt et al., 2018) represent a unique subclass of adversarial attacks that focus on generating localized perturbations to fool deep learning models. Unlike traditional adversarial attacks, which often involve slight pixel-level modifications across the entire image, adversarial patches are confined to small regions but can cause significant misclassifications even when covering only a fraction of the input. This adversarial patch is proven to have more practicality (Athalye et al., 2018), contributing to a deeper understanding of the interaction between digital perturbations and physical environments. Some works (Liu et al., 2016) also explore the transferability of adversarial patches across different models. Concurrently, (Duan et al., 2020) focused on generating adversarial patches using generative models, enhancing the efficiency and effectiveness of attack generation. However, there is still no adversarial patch attack being investigated in LVLM applications.

## 3 THE PROPOSED ATTACK

In this section, we first describe the preliminary adversarial attacks on Large Vision-Language Models (LVLMs). We then present the overview of the proposed attack approach *HardPatch* and illustrate details of each component.

### 3.1 PRELIMINARY

Given the input image $\boldsymbol{x}$ and the input prompt $\boldsymbol{c}_{in}$, an image-grounded text generative LVLM $f_{\Theta}(\boldsymbol{x}, \boldsymbol{c}_{in}) \mapsto \boldsymbol{c}_{out}$ predicts a suitable textual response $\boldsymbol{c}_{out}$, where $\Theta$ is the LVLM's parameters. Since LVLM drivers multiple tasks, in image captioning tasks, for instance, $\boldsymbol{c}_{in}$ is a placeholder $\oslash$ and $\boldsymbol{c}_{out}$ is the caption; in visual question answering tasks, $\boldsymbol{c}_{in}$ is the question and $\boldsymbol{c}_{out}$ is the answer. The adversary typically adds an imperceptible visual perturbation on the benign image to craft an adversarial example $\boldsymbol{x}'$ that misleads the LVLM model $f_{\Theta}$ to output a wrong prediction with a specific prompt $\boldsymbol{c}_{in}$ as:

$$f_{\Theta}(\boldsymbol{x}', \boldsymbol{c}_{in}) \neq f_{\Theta}(\boldsymbol{x}, \boldsymbol{c}_{in}), \text{ s.t. } ||\boldsymbol{x}' - \boldsymbol{x}||_p < \epsilon, \tag{1}$$

where $\epsilon$ is the image perturbation magnitude. Specifically, for the untargeted attack, the attack is successful if the model is misled to generate text different from the prediction with the clean image. For the targeted attack, the attack is considered to be successful only if the prediction exactly matches the attackers' preset target text $\boldsymbol{c}'_{out}$ where $\boldsymbol{c}'_{out} \neq \boldsymbol{c}_{out}$.

In this paper, we focus on the task of hard-label LVLM adversarial attack, *i.e.*, attackers can only access to the predicted text output from the victim LVLM model to generate adversarial examples.

### 3.2 OVERVIEW OF OUR *HardPatch* ATTACK

**Discussion on Our Motivation.** Existing LVLM attackers (Dong et al., 2023; Wang et al., 2023; 2024; Zhang et al., 2024; Luo et al., 2024; Zhao et al., 2024) generally add pixel-wise noise on the whole image input, which are easily optimized in the white-/gray-box or transfer-based black-box

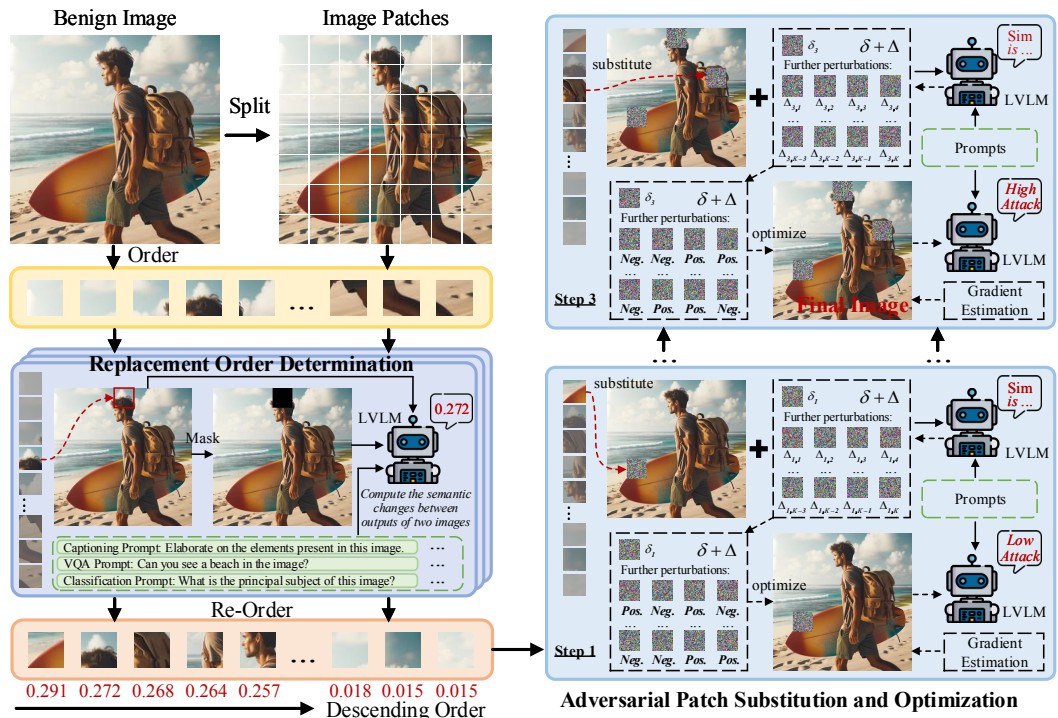

Figure 2: Overview of our proposed *HardPatch* attack. Given the input image and prompts, we first uniformly split the image into patches of the same size. Then, we individually mask each patch to assess their sensitivity to the LVLM model by measuring the semantic changes between their text output with the clean one. After that, we iteratively substitute the most vulnerable patch with noise and estimate gradients to update its noisy pattern. Multiple patches are perturbed until the final altered image achieves the adversarial condition.

setting via the backpropagated gradient. However, in more challenging hard-label setting, it is difficult to directly determine and tamper the LVLM's adversarial attention to optimize previous global noise by solely querying the LVLM model. Inspired by the global semantic invariant characteristic with local contexts mask of MAE (He et al., 2022), we propose to develop attack based on adversarial patch, which assesses the LVLM's vulnerability on local alteration by individually masking different patches of the original images. Then, the patches that have a greater adversarial impact on the LVLM model will be further combined to jointly be perturbed for achieving attacks.

**Overall of Our Attack Pipeline.** The overall pipeline of our *HardPatch* is illustrated in Figure 2. A placement order determination module is first introduced to assess the sensitivity of each patch to the LVLM and re-order the patches. Then, the adversarial patch substitution and optimization module is proposed to alter the patches following the order step-by-step. Multiple patches are perturbed until the attack succeeds. We will provide more details of these two modules in the following.

## 3.3 REPLACEMENT ORDER DETERMINATION OF ADVERSARIAL PATCHES

As for initialization, we first uniformly split the image $x$ into $M$ patches $\{v_1, v_2, ..., v_M\}$. Then, we propose to individually mask each original patch to assess the impact of the corresponding altered sample, where the larger the impact, the more sensitive the LVLM model is to altering the corresponding patch. Therefore, more important patches with greater impact on the victim model should be substituted with noisy patterns at the beginning in the adversarial replacement order. Specifically, to evaluate the importance/sensitivity of each patch $v_m, m \in M$, we set the patch $v_m$ to be all zero and feed the image into the LVLM model. We utilize a lightweight textual encoder (*i.e.*, CLIP (Radford et al., 2021)) to evaluate the semantic similarity between its text output and the clean output as:

$$S(v_m) = \text{Sim}(f_\Theta(x'(v_m), c_{in}), f_\Theta(x, c_{in})), \tag{2}$$

where $\boldsymbol{x}'(\boldsymbol{v}_m)$ denotes generating adversarial sample by altering patch $\boldsymbol{v}_m$, $\mathrm{Sim}(\cdot)$ is the text-aware cosine similarity function and its range is between $[0, 1]$. Then we compute the importance score of each $\boldsymbol{v}_m$ by evaluating the semantic changes by altering patch $\boldsymbol{v}_m$, the large score indicates the better attack performance:

$$\mathcal{I}(\boldsymbol{v}_m) = 1 - \mathcal{S}(\boldsymbol{v}_m). \tag{3}$$

Based on all importance scores $\{\mathcal{I}(\boldsymbol{v}_m)\}_{m=1}^{m=M}$, we sort all patches in descending order as the adversarial replacement order $\mathcal{O} = \{\boldsymbol{v}'_1, \boldsymbol{v}'_2, ..., \boldsymbol{v}'_M\}$ for latter process.

### 3.4 ADVERSARIAL PATCH SUBSTITUTION AND OPTIMIZATION

To achieve hard-label LVLM attack, according to the replacement order $\mathcal{O} = \{\boldsymbol{v}'_1, \boldsymbol{v}'_2, ..., \boldsymbol{v}'_M\}$, we propose to constantly substitute and optimize the most vulnerable patches to query the model for investigating whether the alter can change the output semantics. Beginning at the first patch $\boldsymbol{v}'_1$, we first randomly sample patch-wise noise $\boldsymbol{\delta}_1$ from a uniform distribution to substitute $\boldsymbol{v}'_1$ in the image $\boldsymbol{x}$, then conduct $T$-step gradient estimation to update $\boldsymbol{\delta}_1$ by solely querying the LVLM model. If the $T$-times updated $\boldsymbol{\delta}_1$ can not achieve significant attack performance, we additionally substitute and optimize the latter patch with the same process. The whole attacking procedure of adversarial patch substitution and optimization does not end until the adversarial condition is achieved.

In particular, as for the $m$-th order patch $\boldsymbol{v}'_m$, patch-wise noise $\boldsymbol{\delta}_m$ is initialized to substitute $\boldsymbol{v}'_m$ and we can further optimize it with a reasonable direction by querying the LVLM with additive random noise. Specifically, we first employ a normalized uniform distribution $\boldsymbol{u} \cdot \exp(\boldsymbol{u} - 1)$, $\boldsymbol{u} \sim \mathcal{U}(-1, 1)$ to add a set of slight perturbations $\{\boldsymbol{\Delta}_k\}_{k=1}^{k=K}$ on the patch $\boldsymbol{\delta}_m$ for further altering. At the $t$-th step, we define an indicator function $\varphi_k$ to measure whether the perturbation $\boldsymbol{\Delta}_k$ can cause the misprediction of LVLM model as:

$$\varphi_k^{Tar} = \begin{cases} 1, & \text{If } \mathrm{Sim}(f_\Theta(\boldsymbol{x}'_m(\boldsymbol{\delta}_m + \boldsymbol{\Delta}_k), \boldsymbol{c}_{in}), \boldsymbol{c}'_{out}) > \mathrm{Sim}(f_\Theta(\boldsymbol{x}'_m(\boldsymbol{\delta}_m), \boldsymbol{c}_{in}), \boldsymbol{c}'_{out}), \\ 0, & \text{If } \mathrm{Sim}(f_\Theta(\boldsymbol{x}'_m(\boldsymbol{\delta}_m + \boldsymbol{\Delta}_k), \boldsymbol{c}_{in}), \boldsymbol{c}'_{out}) \leq \mathrm{Sim}(f_\Theta(\boldsymbol{x}'_m(\boldsymbol{\delta}_m), \boldsymbol{c}_{in}), \boldsymbol{c}'_{out}), \end{cases} \tag{4}$$

$$\varphi_k^{Untar} = \begin{cases} 1, & \text{If } \mathrm{Sim}(f_\Theta(\boldsymbol{x}'_m(\boldsymbol{\delta}_m + \boldsymbol{\Delta}_k), \boldsymbol{c}_{in}), \boldsymbol{c}_{out}) < \mathrm{Sim}(f_\Theta(\boldsymbol{x}'_m(\boldsymbol{\delta}_m), \boldsymbol{c}_{in}), \boldsymbol{c}_{out}), \\ 0, & \text{If } \mathrm{Sim}(f_\Theta(\boldsymbol{x}'_m(\boldsymbol{\delta}_m + \boldsymbol{\Delta}_k), \boldsymbol{c}_{in}), \boldsymbol{c}_{out}) \geq \mathrm{Sim}(f_\Theta(\boldsymbol{x}'_m(\boldsymbol{\delta}_m), \boldsymbol{c}_{in}), \boldsymbol{c}_{out}), \end{cases} \tag{5}$$

where $Tar, Untar$ denote the targeted and untargeted attacks, $\boldsymbol{x}'_m$ denotes the image already being substituted by previous patch-wise perturbations with $\{\boldsymbol{\delta}_1 + \boldsymbol{\Delta}, \boldsymbol{\delta}_2 + \boldsymbol{\Delta}, ..., \boldsymbol{\delta}_{m-1} + \boldsymbol{\Delta}\}$. Therefore, following the traditional Monte Carlo method (James, 1980), we estimate the final updating direction

---

**Algorithm 1:** Algorithm of The Proposed Attack

**Input:** Image input $\boldsymbol{x}$, text input $\boldsymbol{c}_{in}$, LVLM model $f_\Theta(\cdot)$
**Output:** Adversarial image with perturbed patches

1 Split image $\boldsymbol{x}$ into $M$ Patches;
2 **for** *each patch $\boldsymbol{v}_m$ in $\boldsymbol{x}$* **do**     // Replacement Order Determination
3  | Compute the importance score $\mathcal{I}(\boldsymbol{v}_m)$ via Eq. (2),(3);
4 **end**
5 Sort all patches based on their importance scores in descending order;
6 **for** *each patch in replacement order* **do** // Adversarial Patch Substitution and Optimization
7  | Replace patch $\boldsymbol{v}'_m$ with initial noise $\boldsymbol{\delta}_m$ on the image $\boldsymbol{x}_m$;
8  | **for** $t = 1 : T$ **do**
9  |  | Optimize $\boldsymbol{\delta}_m$ with a set of slight perturbations $\{\boldsymbol{\Delta}_k\}_{k=1}^{k=K}$ via Eq. (4),(5),(6);
10  | **end**
11  | **if** *adversarial condition is satisfied (i.e., $Sim^{Tar} > \tau_1$ or $Sim^{Untar} < \tau_2$) or Adversarial patch number reaches preset Maximum* **then**
12  |  | break;
13  | **end**
14 **end**
15 **return** The final $\boldsymbol{x}'_m$ is the adversarial sample

---

Table 1: Attack performance on different LVLM models on MS-COCO dataset (Lin et al., 2014). As for targeted attack ($\uparrow$), we report the semantic similarity scores between the LVLM's output and the attackers' chosen label "Unknown". As for untargeted attack ($\downarrow$), we report the semantic similarity scores between the LVLM's output and clean output. More results are in Appendix A.1.

| LVLM Model | Attack Method | Classification | Captioning | VQA | Overall |
|---|---|---|---|---|---|
| BLIP-2 (Li et al., 2023) | Clean$^{Tar}$ | 0.409 | 0.436 | 0.447 | 0.431 |
| | $HardPatch^{Tar}$ | **0.862** | **0.833** | **0.827** | **0.841** |
| | Clean$^{Untar}$ | 1.000 | 1.000 | 1.000 | 1.000 |
| | $HardPatch^{Untar}$ | **0.524** | **0.601** | **0.547** | **0.557** |
| MiniGPT-4 (Zhu et al., 2023) | Clean$^{Tar}$ | 0.438 | 0.451 | 0.463 | 0.450 |
| | $HardPatch^{Tar}$ | **0.849** | **0.815** | **0.872** | **0.845** |
| | Clean$^{Untar}$ | 1.000 | 1.000 | 1.000 | 1.000 |
| | $HardPatch^{Untar}$ | **0.493** | **0.596** | **0.524** | **0.538** |
| LLaVA-1.5 (Liu et al., 2024a) | Clean$^{Tar}$ | 0.385 | 0.479 | 0.436 | 0.433 |
| | $HardPatch^{Tar}$ | **0.875** | **0.841** | **0.880** | **0.865** |
| | Clean$^{Untar}$ | 1.000 | 1.000 | 1.000 | 1.000 |
| | $HardPatch^{Untar}$ | **0.502** | **0.574** | **0.557** | **0.544** |
| InstructBLIP (Dai et al., 2024) | Clean$^{Tar}$ | 0.473 | 0.512 | 0.508 | 0.498 |
| | $HardPatch^{Tar}$ | **0.839** | **0.803** | **0.844** | **0.829** |
| | Clean$^{Untar}$ | 1.000 | 1.000 | 1.000 | 1.000 |
| | $HardPatch^{Untar}$ | **0.510** | **0.565** | **0.526** | **0.534** |

Table 2: Performance comparison ($\uparrow$) with other LVLM attack on ImageNet (Deng et al., 2009).

| Attack | BLIP-2 (Li et al., 2023) | MiniGPT-4 (Zhu et al., 2023) | LLaVA-1.5 Liu et al. (2024a) |
|---|---|---|---|
| Clean (Zhao et al., 2024) | 0.503 | 0.470 | 0.437 |
| MF-it (Zhao et al., 2024) | 0.546 | 0.484 | 0.452 |
| MF-ii (Zhao et al., 2024) | 0.592 | 0.572 | 0.450 |
| MF-ii+it (Zhao et al., 2024) | 0.665 | 0.666 | 0.597 |
| **Ours** | **0.835** | **0.859** | **0.831** |

by weighted averaging over the $K$ possible directions $\{\boldsymbol{\Delta}_k\}_{k=1}^{k=K}$, and optimize $\boldsymbol{\delta}_m$ as:

$$\boldsymbol{\delta}'_m = \boldsymbol{\delta}_m + \frac{\frac{1}{K}\sum_{k=1}^{K}\varphi_k\boldsymbol{\Delta}_k}{||\frac{1}{K}\sum_{k=1}^{K}\varphi_k\boldsymbol{\Delta}_k||_2}. \qquad (6)$$

By iteratively substituting and optimizing each patch with a set of perturbations with $T$-step, we can generate harmful noise with a certain number of perturbed patches to mislead the LVLM model. The overall algorithm of our attack process is summarized in Algorithm 1.

## 4 EXPERIMENTS

### 4.1 EXPERIMENTAL SETUPS

**LVLM Models and Datasets.** To assess the LVLMs' robustness against our attack, We consider four open-source and advanced LVLM models as our evaluation benchmark, including BLIP-2 (Li et al., 2023), MiniGPT-4 (Zhu et al., 2023), LLaVA-1.5 (Liu et al., 2024a), and InstructBLIP (Dai et al., 2024). As for LVLM datasets, we consider three datasets, *i.e.*, MS-COCO (Lin et al., 2014), ImageNet (Deng et al., 2009), and DALL-E (Ramesh et al., 2021; 2022) with tasks of image classification, image captioning, and visual question answering (VQA). Specifically, We follow previous work (Luo et al., 2024) and (Zhao et al., 2024) to construct MS-COCO and ImageNet datasets, respectively. The DALL-E dataset employs a generative method, using random textual descriptions extracted from MS-COCO captions as prompts for image generation powered by GPT-4 (Achiam et al., 2023). Additionally, it includes randomly generated QA pairs based on the images.

**Implementation Details.** For each input image, the patch number $M$ is set to 49. We follow previous work (Zhao et al., 2024) to employ the CLIP model (Radford et al., 2021) to evaluate the semantic similarity in Eq. (2). The optimization number $T$ for each patch is set to 100, and the additive noise number $K$ is set to 200. As for the adversarial condition, the similarity threshold $\tau_1$ for the targeted attack is set to 0.8, and the similarity threshold $\tau_2$ for the untargeted attack is set to

Table 3: Targeted attack performance (↑) of our *HardPatch* on different LVLM models on MS-COCO dataset (Lin et al., 2014) with different target texts. More results are in Appendix A.1.

| Target Text | LVLM Model | Classification | Captioning | VQA | Overall |
|---|---|---|---|---|---|
| I am sorry | BLIP-2 (Li et al., 2023) | 0.801 | 0.792 | 0.837 | 0.811 |
| | MiniGPT-4 (Zhu et al., 2023) | 0.850 | 0.834 | 0.817 | 0.834 |
| | LLaVA-1.5 (Liu et al., 2024a) | 0.862 | 0.841 | 0.874 | 0.859 |
| | InstructBLIP (Dai et al., 2024) | 0.834 | 0.803 | 0.825 | 0.820 |
| I do not know | BLIP-2 (Li et al., 2023) | 0.878 | 0.775 | 0.861 | 0.838 |
| | MiniGPT-4 (Zhu et al., 2023) | 0.825 | 0.809 | 0.842 | 0.825 |
| | LLaVA-1.5 (Liu et al., 2024a) | 0.857 | 0.825 | 0.853 | 0.845 |
| | InstructBLIP (Dai et al., 2024) | 0.836 | 0.799 | 0.828 | 0.821 |
| I cannot answer | BLIP-2 (Li et al., 2023) | 0.843 | 0.816 | 0.839 | 0.833 |
| | MiniGPT-4 (Zhu et al., 2023) | 0.864 | 0.827 | 0.848 | 0.846 |
| | LLaVA-1.5 (Liu et al., 2024a) | 0.872 | 0.824 | 0.866 | 0.854 |
| | InstructBLIP (Dai et al., 2024) | 0.821 | 0.790 | 0.809 | 0.807 |
| Bomb | BLIP-2 (Li et al., 2023) | 0.835 | 0.804 | 0.851 | 0.830 |
| | MiniGPT-4 (Zhu et al., 2023) | 0.819 | 0.843 | 0.820 | 0.827 |
| | LLaVA-1.5 (Liu et al., 2024a) | 0.830 | 0.798 | 0.842 | 0.823 |
| | InstructBLIP (Dai et al., 2024) | 0.806 | 0.782 | 0.815 | 0.801 |

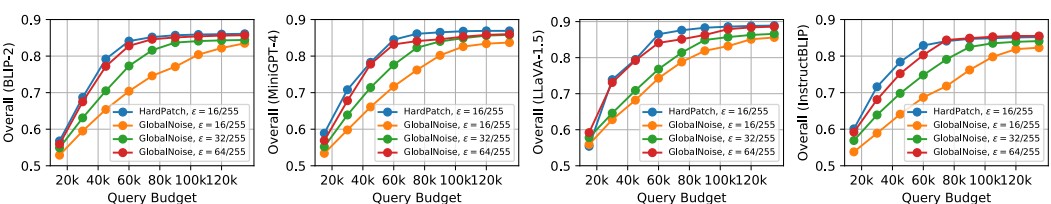

Figure 3: Performance comparison between our adversarial patch and the global noise. Experiments are conducted on four LVLM models on the MS-COCO dataset (Lin et al., 2014).

0.6. The preset maximum adversarial patch number is 4. We impose $\epsilon = 16/255$ as the constraint for. All experiments are conducted on eight NVIDIA H100 Tensor Core GPUs.

## 4.2 ATTACK PERFORMANCE ON TARGETED/UNTARGETED SETTING

To evaluate the effectiveness of the proposed *HardPatch* attack, we show attack performance on different LVLM models on MS-COCO dataset in Table 1. Here, we implement our *HardPatch* in both targeted and untargeted attack settings. As for the targeted attack, we report the semantic similarities between the LVLM's output and the attackers' chosen label, where the larger score denotes better performance. We select the target text "unknown" to avoid the inclusion of high-frequency responses commonly found in vision-language tasks. As for the untargeted attack, we report the semantic similarities between the LVLM's output and clean output, where the smaller score denotes better performance. From this table, we can conclude that: (1) As for the targeted attack, the output of clean images $\text{Clean}^{Tar}$ shares low textual semantic similarity with the target text. By only querying the LVLM model, our $\text{HardPatch}^{Tar}$ can significantly guide the model's output to fit the target text with much higher similarity. (2) As for the untargeted attack, our $\text{HardPatch}^{Untar}$ can keep the model's output away from the clean output with much smaller similarity. We also compare our attack with previous LVLM attacker MF (Zhao et al., 2024) on the same ImageNet (Deng et al., 2009) dataset for fair comparison in Table 2, where our attack still achieves much better performance.

We also extend our evaluation to various other target texts in Table 3. The experiment includes a selection of text with varied length and usage frequency. We can observe that our *HardPatch* attack performs the best overall and in each individual task under different target text, though the similarity differs for different target prompts. In summary, our *HardPatch* can effectively attack the LVLMs in the challenging hard-label setting. More evaluations on other datasets can be found in Appendix A.1

## 4.3 ADVERSARIAL PATCH *vs.* GLOBAL NOISE?

We provide an in-depth analysis of why we should choose the adversarial patch instead of the global noise for attacking hard-label LVLMs. In the hard-label setting, we can not explicitly know how

Table 4: Targeted attack performance (↑) of our *HardPatch* on MS-COCO dataset (Lin et al., 2014) with different maximum adversarial patch number. More results are in Appendix A.4.

| Maximum Number | LVLM Model | Classification | Captioning | VQA | Overall |
|---|---|---|---|---|---|
| Number= 1 | BLIP-2 (Li et al., 2023) | 0.678 | 0.642 | 0.651 | 0.657 |
| | MiniGPT-4 (Zhu et al., 2023) | 0.649 | 0.665 | 0.670 | 0.661 |
| | LLaVA-1.5 (Liu et al., 2024a) | 0.626 | 0.634 | 0.668 | 0.643 |
| | InstructBLIP (Dai et al., 2024) | 0.681 | 0.652 | 0.645 | 0.660 |
| Number= 2 | BLIP-2 (Li et al., 2023) | 0.749 | 0.726 | 0.768 | 0.748 |
| | MiniGPT-4 (Zhu et al., 2023) | 0.761 | 0.704 | 0.753 | 0.739 |
| | LLaVA-1.5 (Liu et al., 2024a) | 0.757 | 0.725 | 0.752 | 0.744 |
| | InstructBLIP (Dai et al., 2024) | 0.772 | 0.730 | 0.746 | 0.750 |
| Number= 3 | BLIP-2 (Li et al., 2023) | 0.822 | 0.804 | 0.800 | 0.809 |
| | MiniGPT-4 (Zhu et al., 2023) | 0.815 | 0.793 | 0.828 | 0.812 |
| | LLaVA-1.5 (Liu et al., 2024a) | 0.861 | 0.807 | 0.836 | 0.835 |
| | InstructBLIP (Dai et al., 2024) | 0.810 | 0.779 | 0.814 | 0.801 |
| Number= 4 | BLIP-2 (Li et al., 2023) | 0.862 | 0.833 | 0.827 | 0.841 |
| | MiniGPT-4 (Zhu et al., 2023) | 0.849 | 0.815 | 0.872 | 0.845 |
| | LLaVA-1.5 (Liu et al., 2024a) | 0.875 | 0.841 | 0.880 | 0.865 |
| | InstructBLIP (Dai et al., 2024) | 0.839 | 0.803 | 0.844 | 0.829 |

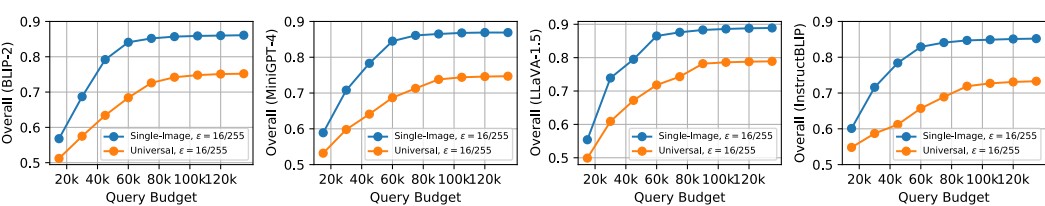

Figure 4: Performance comparison of our *HardPatch* in single-image and universal attack settings. Experiments are conducted on four LVLM models on the MS-COCO dataset (Lin et al., 2014).

LVLM models comprehend and reason the input image according to the prompt. Therefore, without understanding the vulnerability of local image regions, directly adding and optimizing global noise to all pixels of the whole image (using Monte Carlo strategy) makes it difficult to achieve good performance as its optimization/search space is too large and complicated. Unlike this global noise, our *HardPatch* attack is able to implicitly perceive the patch-wise sensitivity to the LVLM model for determining the substitution and optimization location of adversarial patches. We provide detailed experiments on four LVLMs on the MS-COCO dataset in Figure 3. Under the same perturbation budget $\epsilon = 16/255$, global noise requires much more query steps and times (about $2\times$) for optimization, and also achieves relatively worse performance. Although global noise with larger $\epsilon = 64/255$ can achieve similar performance with our method, it significantly increases the noise size, resulting in low-quality and noticeable perturbed images. Therefore, our adversarial patch is more imperceptible and efficient. More experiments and visualizations are illustrated in Appendix A.2.

### 4.4 Extending *HardPatch* to Universal Attack Setting

In all our experiments, we implement our proposed *HardPatch* method in a single-image attack setting, where the perturbed patches vary among different image-text inputs. Further, we can also extend our *HardPatch* attack into a universal attack setting, where the patches are the same among all image-text input. Specifically, we follow the traditional universal setting (Moosavi-Dezfooli et al., 2017) to optimize vulnerable patches. In particular, we first assess the sensitivities of all patches based on their averaged impacts on the whole test set. Then, we jointly optimize the patches in their descending order to attack all image-prompt inputs. As shown in Figure 4, we can conclude that: (1) In the same perturbation budget, the universal attack setting is much more difficult to achieve since different images share diverse sensitive regions in different locations to the LVLM model. Therefore, it requires more querying steps and achieves lower final performance in the targeted attack setting. (2) Instead, the single-image attack is more flexible and can straightforwardly perturb the most vulnerable patches in each image. Therefore, it is more efficient and can achieve better attack performance. More experiments and analysis are provided in Appendix A.3.

Table 5: Targeted attack performance (↑) of our *HardPatch* on MS-COCO dataset (Lin et al., 2014) with different image split. The maximum adversarial patch number is set to 4.

| Image Split $M$ | LVLM Model | Classification | Captioning | VQA | Overall |
|---|---|---|---|---|---|
| Split to $5 \times 5$ | BLIP-2 (Li et al., 2023) | 0.881 | 0.842 | 0.839 | 0.854 |
| | MiniGPT-4 (Zhu et al., 2023) | 0.875 | 0.830 | 0.863 | 0.856 |
| | LLaVA-1.5 (Liu et al., 2024a) | 0.874 | 0.836 | 0.872 | 0.861 |
| | InstructBLIP (Dai et al., 2024) | 0.868 | 0.824 | 0.850 | 0.847 |
| Split to $7 \times 7$ | BLIP-2 (Li et al., 2023) | 0.862 | 0.833 | 0.827 | 0.841 |
| | MiniGPT-4 (Zhu et al., 2023) | 0.849 | 0.815 | 0.872 | 0.845 |
| | LLaVA-1.5 (Liu et al., 2024a) | 0.875 | 0.841 | 0.880 | 0.865 |
| | InstructBLIP (Dai et al., 2024) | 0.839 | 0.803 | 0.844 | 0.829 |
| Split to $9 \times 9$ | BLIP-2 (Li et al., 2023) | 0.849 | 0.821 | 0.816 | 0.828 |
| | MiniGPT-4 (Zhu et al., 2023) | 0.834 | 0.801 | 0.852 | 0.829 |
| | LLaVA-1.5 (Liu et al., 2024a) | 0.861 | 0.829 | 0.870 | 0.853 |
| | InstructBLIP (Dai et al., 2024) | 0.827 | 0.789 | 0.833 | 0.816 |

Table 6: Targeted attack performance (↑) of our *HardPatch* on different patch orders on MS-COCO (Lin et al., 2014) dataset. The maximum adversarial patch number is set to 4.

| Image Split $M$ | LVLM Model | Classification | Captioning | VQA | Overall |
|---|---|---|---|---|---|
| Random Order | BLIP-2 (Li et al., 2023) | 0.714 | 0.697 | 0.680 | 0.697 |
| | MiniGPT-4 (Zhu et al., 2023) | 0.696 | 0.672 | 0.733 | 0.700 |
| | LLaVA-1.5 (Liu et al., 2024a) | 0.729 | 0.703 | 0.737 | 0.723 |
| | InstructBLIP (Dai et al., 2024) | 0.688 | 0.675 | 0.699 | 0.687 |
| Descending Order | BLIP-2 (Li et al., 2023) | 0.862 | 0.833 | 0.827 | 0.841 |
| | MiniGPT-4 (Zhu et al., 2023) | 0.849 | 0.815 | 0.872 | 0.845 |
| | LLaVA-1.5 (Liu et al., 2024a) | 0.875 | 0.841 | 0.880 | 0.865 |
| | InstructBLIP (Dai et al., 2024) | 0.839 | 0.803 | 0.844 | 0.829 |

## 4.5 FURTHER ANALYSIS

**The Influence of the Maximum Number of Adversarial Patches.** The number of adversarial patches is related to the imperceptibility. Therefore, we set a maximum number of adversarial patches during the patch substitution and optimization. To investigate the influence of the maximum number of adversarial patches on the adversarial conditions, we conduct corresponding experiments in Table 4. We can conclude that: (1) Only one adversarial patch is not enough to mask and perturb most images' semantics, resulting in relatively lower attack performance. (2) More adversarial patches can better fool the LVLM model with more vulnerable visual contents. (3) Four adversarial patches are enough to achieve great attack performance. Considering more adversarial patches cost more resources and time, we preset the adversarial patch number to 4 in all our experiments.

**Performance of Attack with Different Image Split.** We also investigate the impact of different settings of image split. In all our experiments, we split each image into $7 \times 7$ patches. As shown in Table 5, we conduct experiments on the image split of $5 \times 5$ and $9 \times 9$, respectively. We can conclude that: Different image splits of the same maximum adversarial patch number share similar attack performances. Since patches in $5 \times 5$ split have more perturbed pixels, it is easier to achieve the attack. Instead, patches in $9 \times 9$ split have fewer perturbed pixels, thus achieving a lower performance. More experiments and analysis are in Appendix A.5.

**Effectiveness of the Replacement Order Determination.** To demonstrate the effectiveness of our proposed module of Replacement Order Determination, we conduct an ablation study in Table 6 where we change our LVLM-sensitive replacement order into a random version. From this table, we can conclude that: (1) Random order may select LVLM's insensitive patches, resulting in more difficult patch optimization for achieving attack. (2) Our Replacement Order Determination can assess the vulnerability of each patch, and provide a descending order for easily achieving attack. Therefore, the proposed Replacement Order Determination module can help efficiently and effectively find the global optimal patches for perturbation.

**Robustness to Defense Strategy.** To evaluate the robustness of our proposed *HardPatch* attack, we follow previous work Luo et al. (2024) to exploit widely used RandomRotation as the defense strategy to defend our generated adversarial examples on four LVLM models. As shown in Table 7,

Table 7: Targeted attack performance ($\uparrow$) of our *HardPatch* against defense strategy of RandomRotation on MS-COCO (Lin et al., 2014) dataset.

| Image Split $M$ | LVLM Model | Classification | Captioning | VQA | Overall |
|---|---|---|---|---|---|
| With Defense | BLIP-2 (Li et al., 2023) | 0.828 | 0.779 | 0.781 | 0.796 |
| | MiniGPT-4 (Zhu et al., 2023) | 0.797 | 0.762 | 0.803 | 0.787 |
| | LLaVA-1.5 (Liu et al., 2024a) | 0.815 | 0.784 | 0.810 | 0.803 |
| | InstructBLIP (Dai et al., 2024) | 0.783 | 0.756 | 0.772 | 0.770 |
| Without Defense | BLIP-2 (Li et al., 2023) | 0.862 | 0.833 | 0.827 | 0.841 |
| | MiniGPT-4 (Zhu et al., 2023) | 0.849 | 0.815 | 0.872 | 0.845 |
| | LLaVA-1.5 (Liu et al., 2024a) | 0.875 | 0.841 | 0.880 | 0.865 |
| | InstructBLIP (Dai et al., 2024) | 0.839 | 0.803 | 0.844 | 0.829 |

Table 8: Analysis on the method complexity of our *HardPatch* attack.

| Module | GPU Hours | GPU Memories |
|---|---|---|
| Replacement Order Determination | 2.4h | 36.2GB |
| Adversarial Patch Substitution and Optimization | 5.6h | 53.8GB |

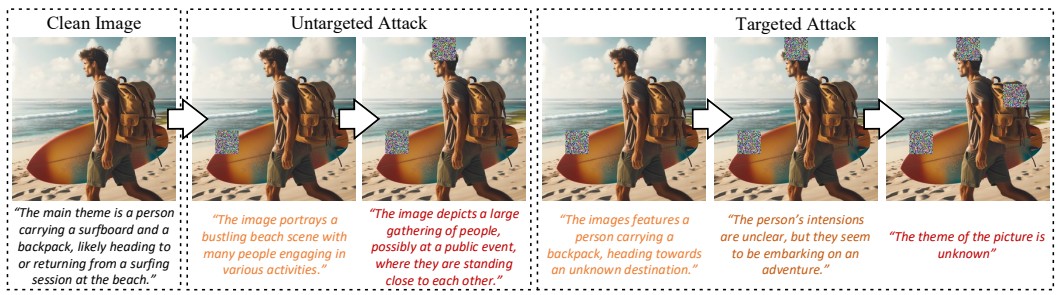

Figure 5: Visualizations on untargeted/targeted adversarial samples and corresponding output for the input prompt "*Convey the main theme of this picture succinctly*" on LLaVA-1.5 (Liu et al., 2024a).

our *HardPatch* just achieves slightly lower performance on the RandomRotation defense, validating that our attack is robust enough against the potential defense strategy.

**Efficiency Analysis.** As shown in Table 8, we provide the GPU hours and memories of generating adversarial examples. We can find that our method is efficient and only costs a few hours for each component. The primary GPU computational and memory overheads occur during the querying stage against the victim LVLM when substituting and optimizing the adversarial patch. This involves adding slight noise to all attack samples during each iterative update of the patch to explore their impacts, and this stage also constitutes the major consumption of the query budget.

**Visualizations.** As shown in Figure 5, we provide visualizations of the step-by-step adversarial examples and corresponding textual output of both untargeted and targeted attacks. We can conclude that the proposed *HardPatch* is effective in fooling the LVLM model by dynamically changing the semantics of original images via adversarial patches. More visualizations are in Appendix A.6.

More experiments, ablation studies, and visualizations can be found in the Appendix.

## 5 CONCLUSION

In this paper, we raise a practical and challenging question, *i.e.*, can visual adversarial patches fool hard-label LVLM models? In particular, we propose the first hard-label adversarial attack method called *HardPatch* against LVLM models by solely querying the input/output of LVLMs. We start by uniformly splitting each image into multiple patches and assessing the vulnerability of LVLMs to different local patches, and then develop a patch substitution and optimization strategy to perturb the most sensitive patches with gradient estimation. Our empirical findings reveal that LVLMs may lose their way when appropriate patches are perturbed. Experiments on a suite of LVLM models and datasets demonstrate the effectiveness of the proposed *HardPatch* attack in the hard-label setting. Future research endeavors will aim at the enhancement of adversarial imperceptibility.

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

Table 9: Attack performance on different LVLM models on more datasets. As for targeted attack ($\uparrow$), we report the semantic similarity scores between the LVLM's output and the attackers' chosen label "Unknown". As for untargeted attack ($\downarrow$), we report the semantic similarity scores between the LVLM's output and clean output.

| LVLM Model | Attack Method | Classification | Captioning | VQA | Overall |
|---|---|---|---|---|---|
| Dataset: ImageNet (Deng et al., 2009) | | | | | |
| BLIP-2 (Li et al., 2023) | $Clean^{Tar}$ | 0.415 | 0.462 | 0.473 | 0.450 |
| | $HardPatch^{Tar}$ | **0.831** | **0.814** | **0.860** | **0.835** |
| | $Clean^{Untar}$ | 1.000 | 1.000 | 1.000 | 1.000 |
| | $HardPatch^{Untar}$ | **0.543** | **0.582** | **0.556** | **0.560** |
| MiniGPT-4 (Zhu et al., 2023) | $Clean^{Tar}$ | 0.419 | 0.447 | 0.504 | 0.457 |
| | $HardPatch^{Tar}$ | **0.837** | **0.862** | **0.879** | **0.859** |
| | $Clean^{Untar}$ | 1.000 | 1.000 | 1.000 | 1.000 |
| | $HardPatch^{Untar}$ | **0.504** | **0.581** | **0.535** | **0.541** |
| LLaVA-1.5 (Liu et al., 2024a) | $Clean^{Tar}$ | 0.448 | 0.434 | 0.459 | 0.447 |
| | $HardPatch^{Tar}$ | **0.826** | **0.803** | **0.865** | **0.831** |
| | $Clean^{Untar}$ | 1.000 | 1.000 | 1.000 | 1.000 |
| | $HardPatch^{Untar}$ | **0.498** | **0.557** | **0.542** | **0.532** |
| InstructBLIP (Dai et al., 2024) | $Clean^{Tar}$ | 0.453 | 0.487 | 0.462 | 0.467 |
| | $HardPatch^{Tar}$ | **0.830** | **0.841** | **0.859** | **0.843** |
| | $Clean^{Untar}$ | 1.000 | 1.000 | 1.000 | 1.000 |
| | $HardPatch^{Untar}$ | **0.522** | **0.568** | **0.544** | **0.545** |
| Dataset: DALL-E (Ramesh et al., 2021; 2022) | | | | | |
| BLIP-2 (Li et al., 2023) | $Clean^{Tar}$ | 0.368 | 0.425 | 0.466 | 0.419 |
| | $HardPatch^{Tar}$ | **0.802** | **0.841** | **0.848** | **0.830** |
| | $Clean^{Untar}$ | 1.000 | 1.000 | 1.000 | 1.000 |
| | $HardPatch^{Untar}$ | **0.539** | **0.594** | **0.525** | **0.553** |
| MiniGPT-4 (Zhu et al., 2023) | $Clean^{Tar}$ | 0.396 | 0.441 | 0.497 | 0.445 |
| | $HardPatch^{Tar}$ | **0.816** | **0.847** | **0.864** | **0.842** |
| | $Clean^{Untar}$ | 1.000 | 1.000 | 1.000 | 1.000 |
| | $HardPatch^{Untar}$ | **0.508** | **0.573** | **0.546** | **0.541** |
| LLaVA-1.5 (Liu et al., 2024a) | $Clean^{Tar}$ | 0.407 | 0.453 | 0.517 | 0.459 |
| | $HardPatch^{Tar}$ | **0.831** | **0.815** | **0.850** | **0.832** |
| | $Clean^{Untar}$ | 1.000 | 1.000 | 1.000 | 1.000 |
| | $HardPatch^{Untar}$ | **0.520** | **0.552** | **0.531** | **0.535** |
| InstructBLIP (Dai et al., 2024) | $Clean^{Tar}$ | 0.434 | 0.469 | 0.483 | 0.462 |
| | $HardPatch^{Tar}$ | **0.823** | **0.874** | **0.836** | **0.844** |
| | $Clean^{Untar}$ | 1.000 | 1.000 | 1.000 | 1.000 |
| | $HardPatch^{Untar}$ | **0.515** | **0.566** | **0.537** | **0.539** |

## A    APPENDIX

In this appendix, we describe additional experiment results and analyses, to support the methods proposed in the main paper.

### A.1    ATTACK PERFORMANCE ON MORE DATASETS

To further demonstrate the effectiveness of the proposed *HardPatch* attack, we show more attack performance on different LVLM models on ImageNet and DALL-E datasets in Table 9. Similar to the experiments in the main paper, we implement our *HardPatch* in both targeted and untargeted attack settings. As for the targeted attack, we report the semantic similarities between the LVLM's output and the attackers' chosen label, where the larger score denotes better performance. We select the target text "unknown" to avoid the inclusion of high-frequency responses commonly found in vision-language tasks. As for the untargeted attack, we report the semantic similarities between the LVLM's output and clean output, where the smaller score denotes better performance. We can conclude that our *HardPatch* can achieve great attack performance in both targeted and untargeted attack settings.

Table 10: Targeted attack performance (↑) of our *HardPatch* on different LVLM models on more datasets with different target texts.

| Target Text | LVLM Model | Classification | Captioning | VQA | Overall |
|---|---|---|---|---|---|
| | Dataset: ImageNet (Deng et al., 2009) | | | | |
| I am sorry | BLIP-2 (Li et al., 2023) | 0.824 | 0.798 | 0.842 | 0.821 |
| | MiniGPT-4 (Zhu et al., 2023) | 0.869 | 0.851 | 0.837 | 0.852 |
| | LLaVA-1.5 (Liu et al., 2024a) | 0.844 | 0.823 | 0.865 | 0.844 |
| | InstructBLIP (Dai et al., 2024) | 0.842 | 0.806 | 0.831 | 0.826 |
| I do not know | BLIP-2 (Li et al., 2023) | 0.853 | 0.790 | 0.837 | 0.827 |
| | MiniGPT-4 (Zhu et al., 2023) | 0.842 | 0.818 | 0.829 | 0.830 |
| | LLaVA-1.5 (Liu et al., 2024a) | 0.836 | 0.825 | 0.841 | 0.834 |
| | InstructBLIP (Dai et al., 2024) | 0.853 | 0.807 | 0.824 | 0.828 |
| I cannot answer | BLIP-2 (Li et al., 2023) | 0.859 | 0.824 | 0.811 | 0.831 |
| | MiniGPT-4 (Zhu et al., 2023) | 0.872 | 0.838 | 0.850 | 0.853 |
| | LLaVA-1.5 (Liu et al., 2024a) | 0.841 | 0.799 | 0.826 | 0.822 |
| | InstructBLIP (Dai et al., 2024) | 0.835 | 0.813 | 0.822 | 0.823 |
| Bomb | BLIP-2 (Li et al., 2023) | 0.833 | 0.797 | 0.854 | 0.828 |
| | MiniGPT-4 (Zhu et al., 2023) | 0.840 | 0.829 | 0.856 | 0.842 |
| | LLaVA-1.5 (Liu et al., 2024a) | 0.831 | 0.805 | 0.844 | 0.827 |
| | InstructBLIP (Dai et al., 2024) | 0.829 | 0.798 | 0.832 | 0.820 |
| | Dataset: DALL-E (Ramesh et al., 2021; 2022) | | | | |
| I am sorry | BLIP-2 (Li et al., 2023) | 0.836 | 0.810 | 0.845 | 0.830 |
| | MiniGPT-4 (Zhu et al., 2023) | 0.848 | 0.821 | 0.859 | 0.843 |
| | LLaVA-1.5 (Liu et al., 2024a) | 0.829 | 0.796 | 0.842 | 0.822 |
| | InstructBLIP (Dai et al., 2024) | 0.857 | 0.824 | 0.833 | 0.838 |
| I do not know | BLIP-2 (Li et al., 2023) | 0.842 | 0.809 | 0.828 | 0.826 |
| | MiniGPT-4 (Zhu et al., 2023) | 0.853 | 0.835 | 0.831 | 0.839 |
| | LLaVA-1.5 (Liu et al., 2024a) | 0.844 | 0.822 | 0.817 | 0.828 |
| | InstructBLIP (Dai et al., 2024) | 0.835 | 0.846 | 0.840 | 0.841 |
| I cannot answer | BLIP-2 (Li et al., 2023) | 0.852 | 0.818 | 0.824 | 0.831 |
| | MiniGPT-4 (Zhu et al., 2023) | 0.861 | 0.843 | 0.837 | 0.847 |
| | LLaVA-1.5 (Liu et al., 2024a) | 0.849 | 0.827 | 0.819 | 0.832 |
| | InstructBLIP (Dai et al., 2024) | 0.836 | 0.834 | 0.832 | 0.834 |
| Bomb | BLIP-2 (Li et al., 2023) | 0.815 | 0.786 | 0.839 | 0.817 |
| | MiniGPT-4 (Zhu et al., 2023) | 0.828 | 0.812 | 0.830 | 0.823 |
| | LLaVA-1.5 (Liu et al., 2024a) | 0.807 | 0.823 | 0.831 | 0.820 |
| | InstructBLIP (Dai et al., 2024) | 0.814 | 0.791 | 0.822 | 0.809 |

To demonstrate that the effectiveness of the proposed *HardPatch* method is not constrained to the specific case of the target text "unknown", we extend our evaluation to various other target texts. The experiment includes a selection of text with varied length and usage frequency. As shown in Table 10, the experiment includes a selection of text with varied length and usage frequency. We can observe that our *HardPatch* attack performs the best overall and in each individual task under different target text, though the similarity differs for different target prompts. In summary, our *HardPatch* can effectively attack the LVLMs in the challenging hard-label setting.

We provide the visualization results of the adversarial examples generated by our *HardPatch* method. As shown in Figure 6, we show the adversarial examples generated by four LVLM models in the targeted setting. we can conclude that: (1) Our *HardPatch* attack can successfully fool these four LVLM models with a smaller number of patches, demonstrating the effectiveness of the proposed method. (2) Different LVLM models have different attention scores on the same patch of the image. Therefore, their generated patches are in different locations. (3) In most cases, two or three patches are enough to fool the victim models. This demonstrates that our patch-based adversarial design is imperceptible.

We also provide the visualization comparison of the adversarial examples generated in targeted and untargeted attack settings. As shown in Figure 7, we can conclude that: (1) Our *HardPatch* attack can successfully fool the LVLM model in both targeted and untargeted settings with a smaller number of patches, demonstrating the effectiveness of the proposed method. (2) The LVLM model has different attention scores on the same patch of different images. Therefore, its generated patches for different images are in different locations. (3) The untargeted attack is much easier to attack than

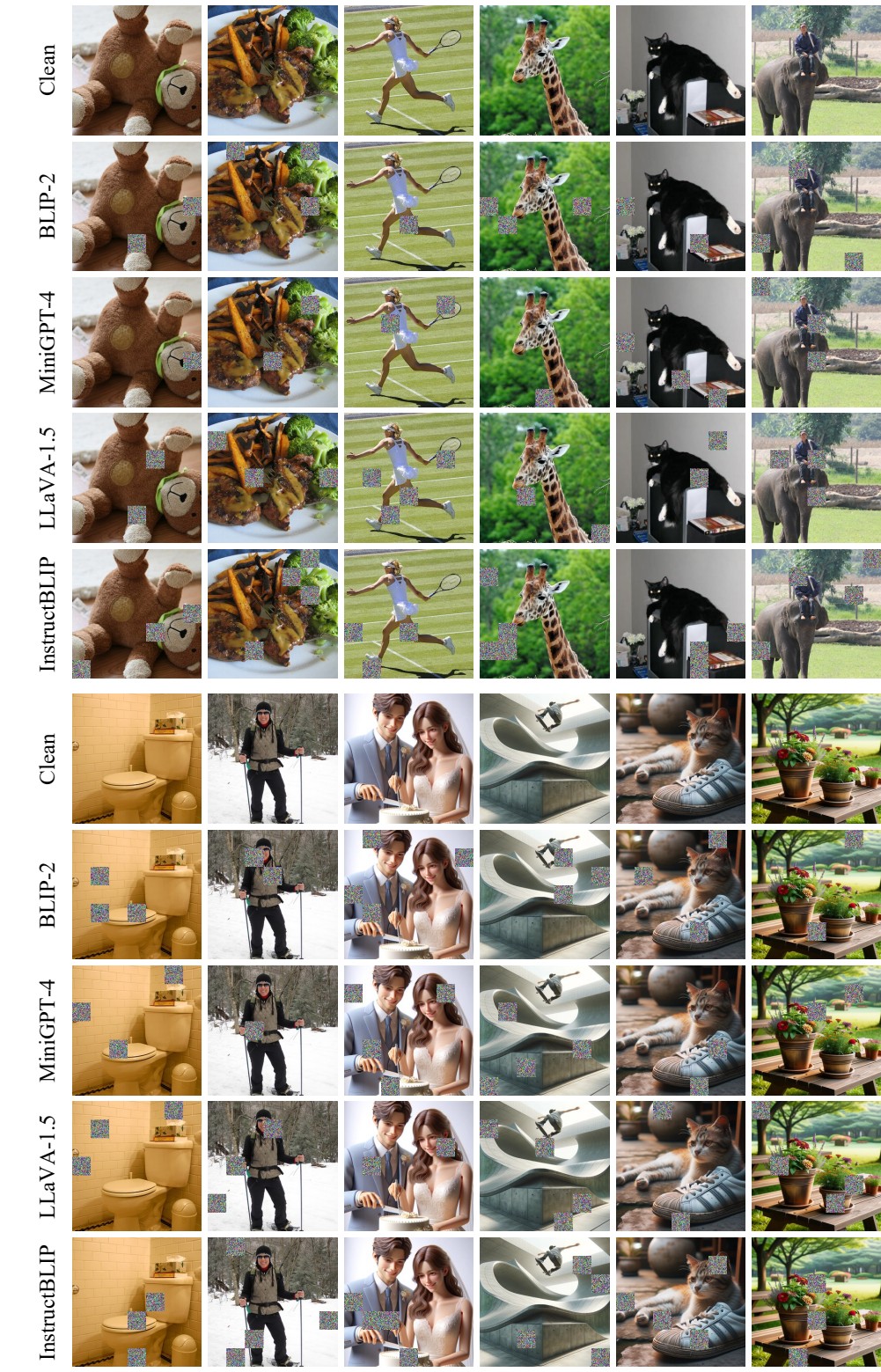

Figure 6: Visualization of the adversarial examples generated with different LVLM models in the targeted attack setting.

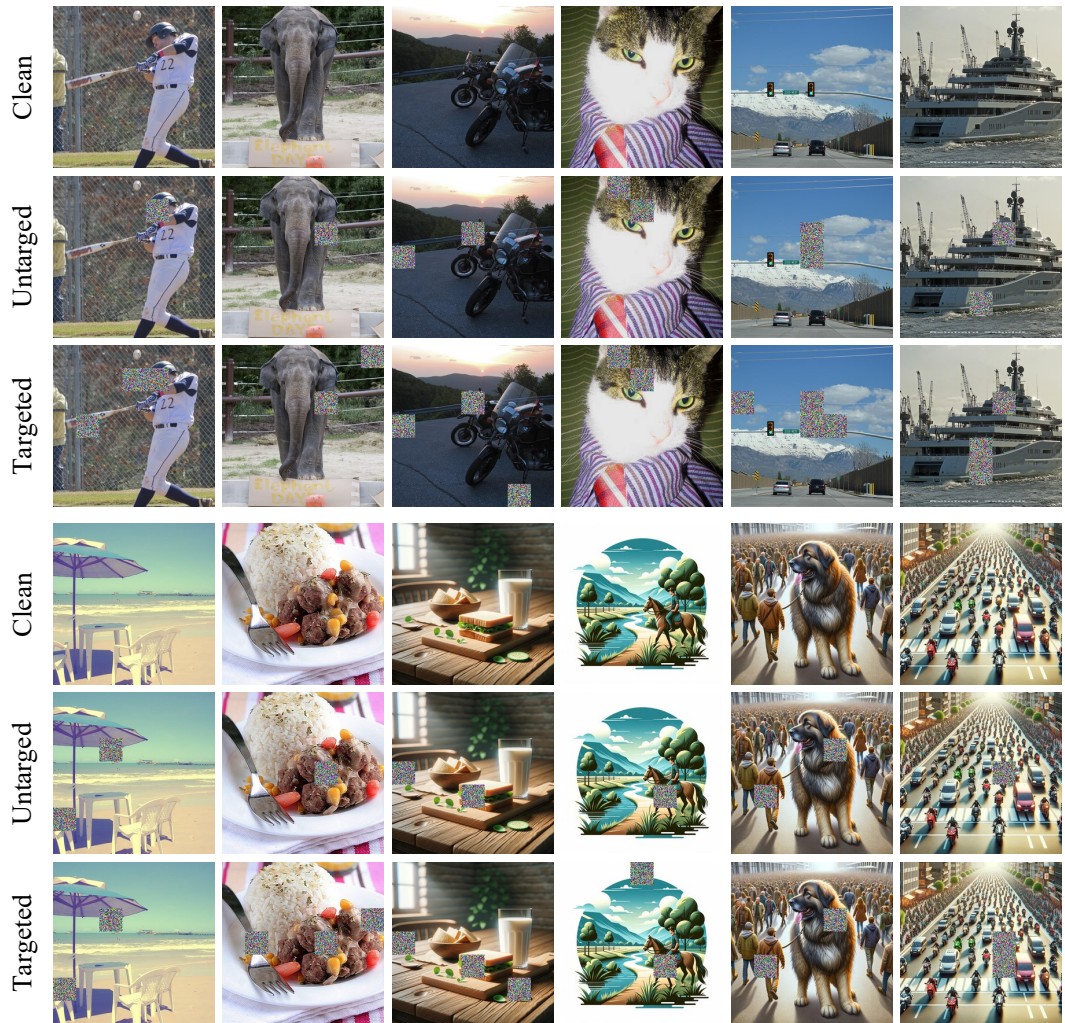

Figure 7: Visualization of the adversarial examples generated with LLaVA-1.5 (Liu et al., 2024a) in both targeted and untargeted attack settings.

the targeted attack, because it only needs to push the output semantic far away from the original one while the targeted attack aims to guide the output semantic to a certain one (which is more difficult). Therefore, the number of adversarial patches is fewer in the untargeted setting.

## A.2 MORE COMPARISONS BETWEEN OUR ADVERSARIAL PATCH AND GLOBAL NOISE

We provide more analysis of why we should choose the adversarial patch instead of the global noise for attacking hard-label LVLMs. Since attackers can not explicitly know how LVLM models comprehend and reason the input image according to the prompt in the hard-label setting, without understanding the vulnerability of local image regions, directly adding and optimizing global noise to all pixels of the whole image (using Monte Carlo strategy) makes it difficult to achieve good performance as its optimization/search space is too large and complicated. Unlike this global noise, our *HardPatch* attack is able to implicitly perceive the patch-wise sensitivity to the LVLM model for determining the substitution and optimization location of adversarial patches. We provide detailed experiments on four LVLMs on ImageNet and DALL-E datasets in Figure 9 and Figure 10. We can conclude that: (1) Under the same perturbation budget $\epsilon = 16/255$, global noise requires much more query steps and times (about $2\times$) for optimization, and also achieves relatively worse performance. (2) Although global noise with larger $\epsilon = 64/255$ can achieve similar performance with our method, it significantly increases the noise size, resulting in low-quality and noticeable perturbed images.

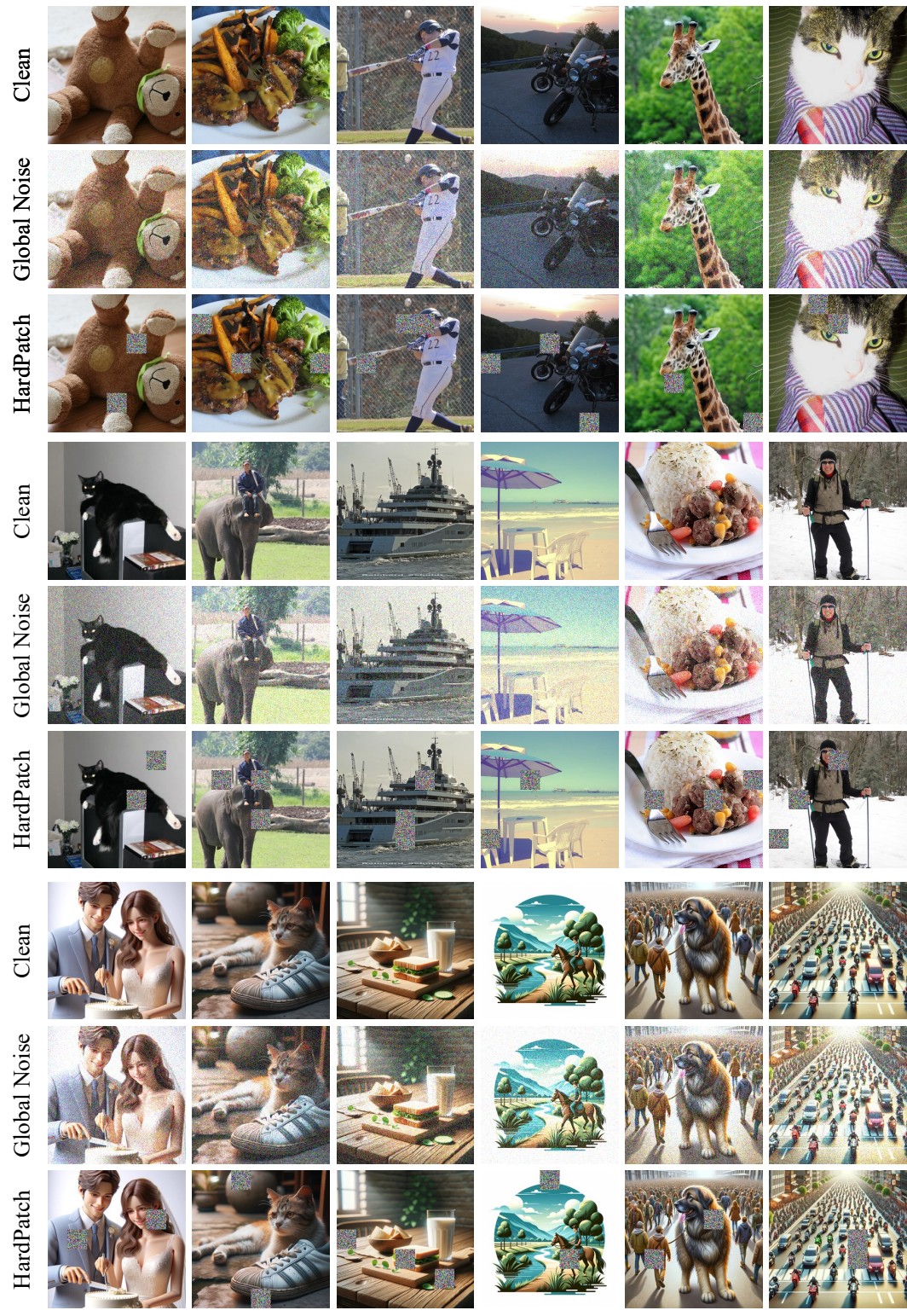

Figure 8: Visualization of the adversarial examples generated by our *HardPatch* and the global noise on LLaVA-1.5 (Liu et al., 2024a) under the targeted attack.

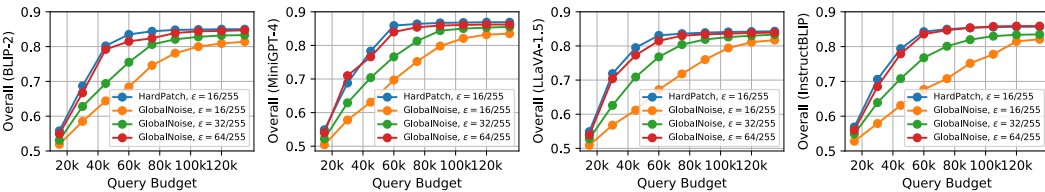

Figure 9: Performance comparison between our adversarial patch and the global noise. Experiments are conducted on four LVLM models on the ImageNet dataset (Deng et al., 2009).

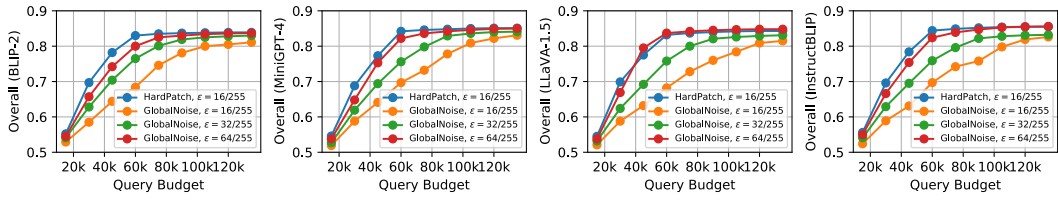

Figure 10: Performance comparison between our adversarial patch and the global noise. Experiments are conducted on four LVLM models on the DALL-E dataset (Ramesh et al., 2021; 2022).

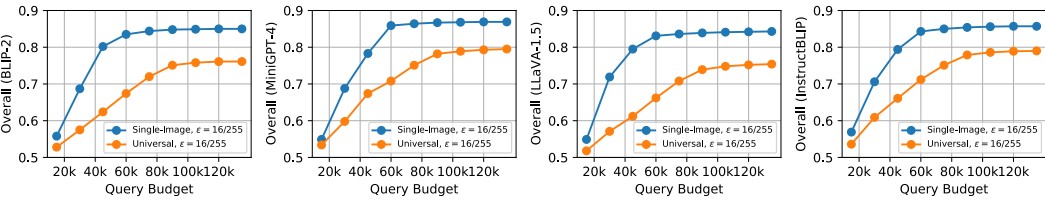

Figure 11: Performance comparison of our *HardPatch* in single-image and universal attack settings. Experiments are conducted on four LVLM models on the ImageNet dataset (Deng et al., 2009).

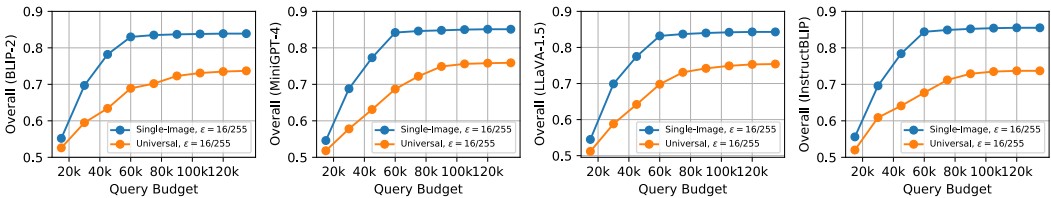

Figure 12: Performance comparison of our *HardPatch* in single-image and universal attack settings. Experiments are conducted on four LVLM models on the DALL-E dataset (Ramesh et al., 2021; 2022).

(3) Our adversarial patch can efficiently be generated to attack the LVLM models with low noise size $\epsilon = 16/255$. We also provide the visualization results of adversarial examples generated by our adversarial patch and global noise in Figure 8. It shows that global noise is very large and noticeable, while our adversarial patch is easier to add to the images and is relatively more imperceptible.

### A.3 MORE EXPERIMENTS ON UNIVERSAL ATTACK SETTING

Our *HardPatch* method is generally implemented in a single-image attack setting, where the perturbed patches vary among different image-text inputs. Further, our *HardPatch* attack can be extended into a universal attack setting, where the patches are optimized to be the same among all image-text input. Specifically, we follow the traditional universal setting (Moosavi-Dezfooli et al., 2017) by first assessing the sensitivities of all patches based on their averaged impacts on the images by querying the LVLM models with different text prompts. Then, we jointly optimize the patches in

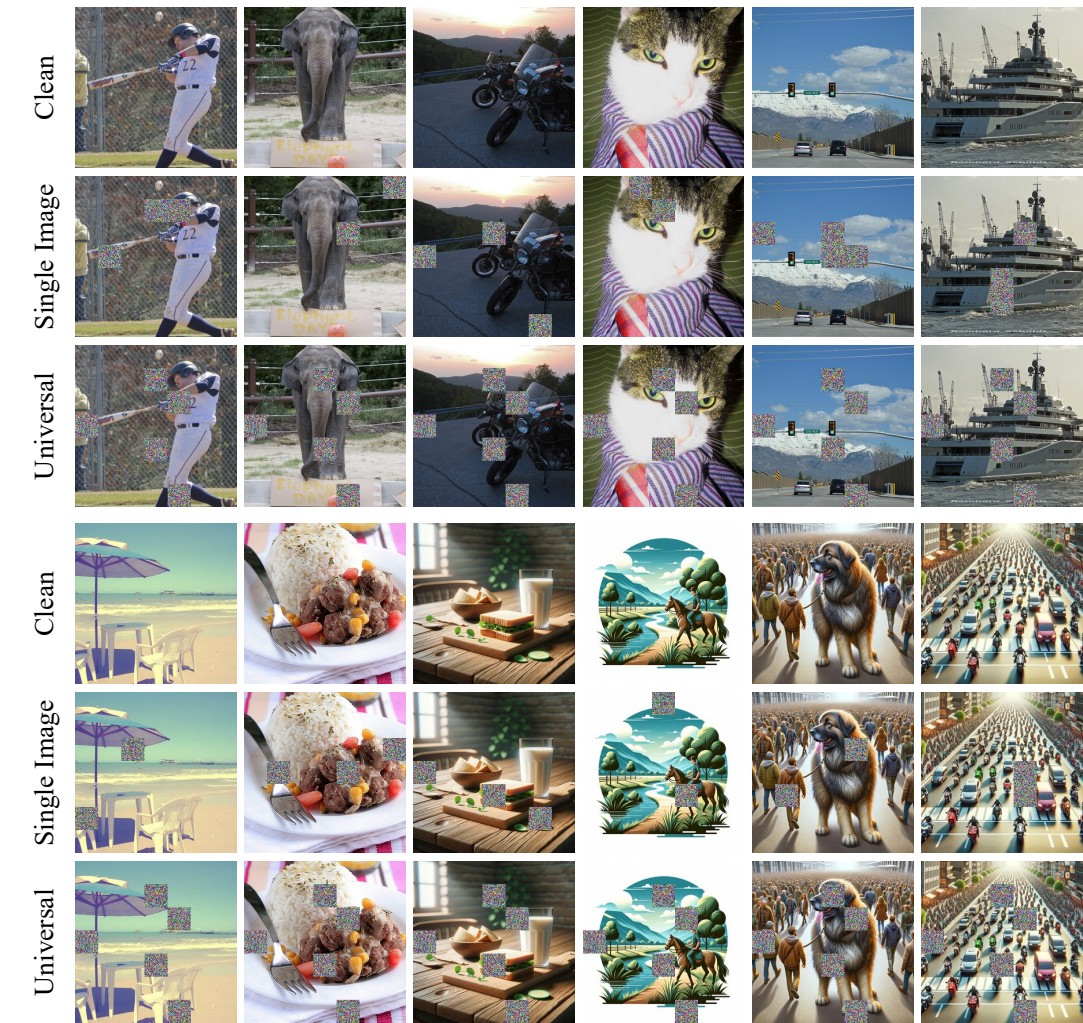

Figure 13: Visualization of the adversarial examples generated with LLaVA-1.5 (Liu et al., 2024a) in both single-image attack and universal attack settings.

their descending order to attack all image-prompt inputs. As shown in Figure 11 and Figure 12, in the same perturbation budget, the single-image attack is more flexible and efficient than the universal attack setting, thus achieving better performance with fewer query budgets. This is because the single-image attack can straightforwardly perturb the most vulnerable patches in each image. Visualization comparisons are further shown in Figure 13, where the universal attack setting is much more difficult to achieve since different images share diverse sensitive regions in different locations to the LVLM model, requiring a larger number of adversarial patches.

## A.4 MORE EXPERIMENTS ON ADVERSARIAL PATCH NUMBER

The number of adversarial patches is related to the imperceptibility. Since more adversarial patches will mask most image contents and lead to noticeable noise (which is also not meaningful), in our attack algorithm, we preset the maximum number of adversarial patches to a fixed number of 4. That means, only $\{1, 2, 3, 4\}$ adversarial patches may be added to the image. To further investigate the influence of the maximum number of adversarial patches on more datasets, we conduct corresponding experiments in Table 11 by presenting different maximum numbers of adversarial patches. We can conclude that: (1) Only one adversarial patch is not enough to mask and perturb most images' semantics, resulting in lower attack performance. (2) More adversarial patches can better fool the LVLM model with more vulnerable visual contents. (3) Four adversarial patches are enough to

Table 11: Targeted attack performance ($\uparrow$) of our *HardPatch* on other datasets with different maximum adversarial patch number.

| Maximum Number | LVLM Model | Classification | Captioning | VQA | Overall |
|---|---|---|---|---|---|
| | Dataset: ImageNet (Deng et al., 2009) | | | | |
| Number= 1 | BLIP-2 (Li et al., 2023) | 0.647 | 0.673 | 0.665 | 0.662 |
| | MiniGPT-4 (Zhu et al., 2023) | 0.668 | 0.638 | 0.654 | 0.653 |
| | LLaVA-1.5 (Liu et al., 2024a) | 0.651 | 0.639 | 0.676 | 0.655 |
| | InstructBLIP (Dai et al., 2024) | 0.642 | 0.640 | 0.657 | 0.646 |
| Number= 2 | BLIP-2 (Li et al., 2023) | 0.773 | 0.749 | 0.758 | 0.760 |
| | MiniGPT-4 (Zhu et al., 2023) | 0.756 | 0.752 | 0.731 | 0.746 |
| | LLaVA-1.5 (Liu et al., 2024a) | 0.752 | 0.725 | 0.739 | 0.738 |
| | InstructBLIP (Dai et al., 2024) | 0.742 | 0.750 | 0.738 | 0.743 |
| Number= 3 | BLIP-2 (Li et al., 2023) | 0.824 | 0.785 | 0.832 | 0.814 |
| | MiniGPT-4 (Zhu et al., 2023) | 0.804 | 0.819 | 0.840 | 0.821 |
| | LLaVA-1.5 (Liu et al., 2024a) | 0.798 | 0.777 | 0.833 | 0.803 |
| | InstructBLIP (Dai et al., 2024) | 0.816 | 0.808 | 0.819 | 0.815 |
| Number= 4 | BLIP-2 (Li et al., 2023) | 0.831 | 0.814 | 0.860 | 0.835 |
| | MiniGPT-4 (Zhu et al., 2023) | 0.837 | 0.862 | 0.879 | 0.859 |
| | LLaVA-1.5 (Liu et al., 2024a) | 0.826 | 0.803 | 0.865 | 0.831 |
| | InstructBLIP (Dai et al., 2024) | 0.830 | 0.841 | 0.859 | 0.843 |
| | Dataset: DALL-E (Ramesh et al., 2021; 2022) | | | | |
| Number= 1 | BLIP-2 (Li et al., 2023) | 0.670 | 0.629 | 0.653 | 0.651 |
| | MiniGPT-4 (Zhu et al., 2023) | 0.625 | 0.664 | 0.652 | 0.647 |
| | LLaVA-1.5 (Liu et al., 2024a) | 0.658 | 0.636 | 0.639 | 0.644 |
| | InstructBLIP (Dai et al., 2024) | 0.643 | 0.649 | 0.680 | 0.657 |
| Number= 2 | BLIP-2 (Li et al., 2023) | 0.764 | 0.728 | 0.751 | 0.748 |
| | MiniGPT-4 (Zhu et al., 2023) | 0.759 | 0.735 | 0.762 | 0.752 |
| | LLaVA-1.5 (Liu et al., 2024a) | 0.738 | 0.716 | 0.747 | 0.734 |
| | InstructBLIP (Dai et al., 2024) | 0.754 | 0.723 | 0.744 | 0.740 |
| Number= 3 | BLIP-2 (Li et al., 2023) | 0.812 | 0.786 | 0.815 | 0.804 |
| | MiniGPT-4 (Zhu et al., 2023) | 0.796 | 0.809 | 0.835 | 0.813 |
| | LLaVA-1.5 (Liu et al., 2024a) | 0.820 | 0.789 | 0.827 | 0.812 |
| | InstructBLIP (Dai et al., 2024) | 0.806 | 0.792 | 0.819 | 0.806 |
| Number= 4 | BLIP-2 (Li et al., 2023) | 0.802 | 0.841 | 0.848 | 0.830 |
| | MiniGPT-4 (Zhu et al., 2023) | 0.816 | 0.847 | 0.864 | 0.842 |
| | LLaVA-1.5 (Liu et al., 2024a) | 0.831 | 0.815 | 0.850 | 0.832 |
| | InstructBLIP (Dai et al., 2024) | 0.823 | 0.874 | 0.836 | 0.844 |

achieve great attack performance. Of course, the adversarial patch number larger than 4 can further boost the attack performance. However, considering more adversarial patches cost more resources and time, we preset the adversarial patch number to 4 in all our experiments.

## A.5 MORE EXPERIMENTS ON IMAGE SPLIT

We also investigate the impact of different settings of image split. In all our experiments, we split each image into $7 \times 7$ patches. As shown in Table 12, we conduct experiments on the image split of $5 \times 5$ and $9 \times 9$, respectively. We can conclude that: Different image splits of the same maximum adversarial patch number share similar attack performances. Since patches in $5 \times 5$ split have more perturbed pixels, it is easier to achieve the attack. Instead, patches in $9 \times 9$ split have fewer perturbed pixels, thus achieving a lower performance. Therefore, we set the split of each image into $7 \times 7$ patches in all our experiments.

## A.6 MORE VISUALIZATION RESULTS

As shown in Figure 14, we provide more visualizations of the step-by-step adversarial examples and corresponding textual output of both untargeted and targeted attacks. We can conclude that the proposed *HardPatch* is effective in fooling the LVLM model by dynamically changing the semantics of original images via adversarial patches.

Table 12: Targeted attack performance (↑) of our *HardPatch* on more datasets with different image split. The maximum adversarial patch number is set to 4.

| Image Split $M$ | LVLM Model | Classification | Captioning | VQA | Overall |
|---|---|---|---|---|---|
| \multicolumn{6}{c}{Dataset: ImageNet (Deng et al., 2009)} | | | | | |
| Split to $5 \times 5$ | BLIP-2 (Li et al., 2023) | 0.842 | 0.826 | 0.853 | 0.840 |
| | MiniGPT-4 (Zhu et al., 2023) | 0.834 | 0.870 | 0.867 | 0.857 |
| | LLaVA-1.5 (Liu et al., 2024a) | 0.839 | 0.831 | 0.855 | 0.842 |
| | InstructBLIP (Dai et al., 2024) | 0.858 | 0.815 | 0.872 | 0.848 |
| Split to $7 \times 7$ | BLIP-2 (Li et al., 2023) | 0.831 | 0.814 | 0.860 | 0.835 |
| | MiniGPT-4 (Zhu et al., 2023) | 0.837 | 0.862 | 0.879 | 0.859 |
| | LLaVA-1.5 (Liu et al., 2024a) | 0.826 | 0.803 | 0.865 | 0.831 |
| | InstructBLIP (Dai et al., 2024) | 0.830 | 0.841 | 0.859 | 0.843 |
| Split to $9 \times 9$ | BLIP-2 (Li et al., 2023) | 0.822 | 0.801 | 0.844 | 0.822 |
| | MiniGPT-4 (Zhu et al., 2023) | 0.830 | 0.819 | 0.847 | 0.832 |
| | LLaVA-1.5 (Liu et al., 2024a) | 0.815 | 0.782 | 0.838 | 0.812 |
| | InstructBLIP (Dai et al., 2024) | 0.814 | 0.813 | 0.836 | 0.821 |
| \multicolumn{6}{c}{Dataset: DALL-E (Ramesh et al., 2021; 2022)} | | | | | |
| Split to $5 \times 5$ | BLIP-2 (Li et al., 2023) | 0.837 | 0.829 | 0.841 | 0.836 |
| | MiniGPT-4 (Zhu et al., 2023) | 0.829 | 0.832 | 0.866 | 0.842 |
| | LLaVA-1.5 (Liu et al., 2024a) | 0.848 | 0.820 | 0.853 | 0.840 |
| | InstructBLIP (Dai et al., 2024) | 0.842 | 0.853 | 0.860 | 0.851 |
| Split to $7 \times 7$ | BLIP-2 (Li et al., 2023) | 0.802 | 0.841 | 0.848 | 0.830 |
| | MiniGPT-4 (Zhu et al., 2023) | 0.816 | 0.847 | 0.864 | 0.842 |
| | LLaVA-1.5 (Liu et al., 2024a) | 0.831 | 0.815 | 0.850 | 0.832 |
| | InstructBLIP (Dai et al., 2024) | 0.823 | 0.874 | 0.836 | 0.844 |
| Split to $9 \times 9$ | BLIP-2 (Li et al., 2023) | 0.814 | 0.838 | 0.832 | 0.828 |
| | MiniGPT-4 (Zhu et al., 2023) | 0.815 | 0.843 | 0.859 | 0.839 |
| | LLaVA-1.5 (Liu et al., 2024a) | 0.820 | 0.799 | 0.827 | 0.815 |
| | InstructBLIP (Dai et al., 2024) | 0.809 | 0.852 | 0.825 | 0.829 |

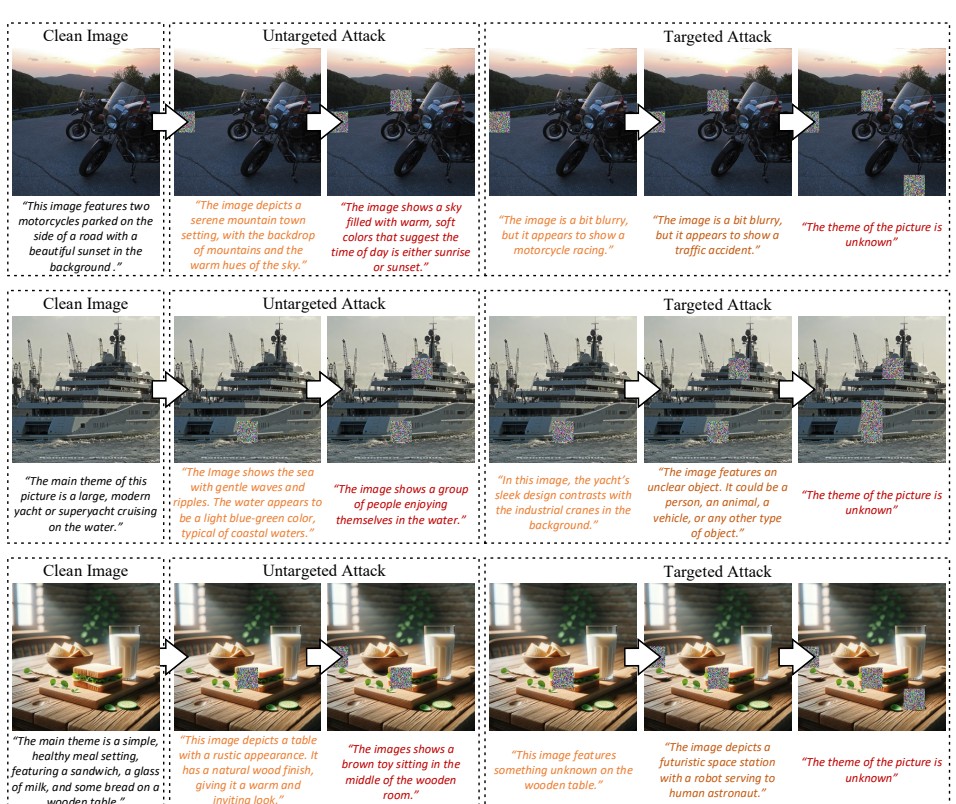

Figure 14: Visualizations on untargeted/targeted adversarial samples and corresponding output.

## A.7 VISUALIZATION ON THE VULNERABILITY OF DIFFERENT PATCHES

At last, we visualize the sensitive scores of different patches of the same images to the LVLM model as shown in Figure 15. Here, the image is divided into $7 \times 7$ patches, and the sensitive score of each patch is measured by the semantic changes between the original output and the output of masking the corresponding patch. The heatmap of each image is computed by further using a softmax function on the scores of whole patches. From this figure, we can conclude that: (1) Different LVLM models have different attentions on different patches of the same image. (2) Masking patches provide a promising way to measure the vulnerability of the LVLM models to the local regions of input images. Based on the sensitivity scores of different patches, researchers can design specific local perturbations for attacking the LVLM models.

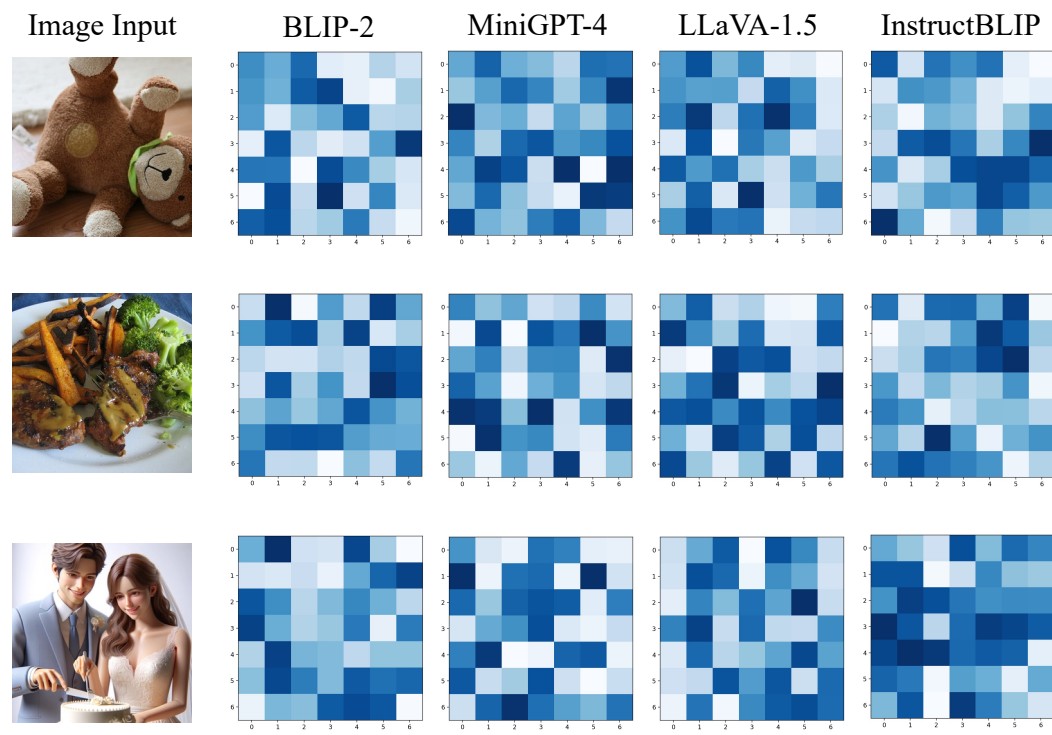

Figure 15: Visualizations on the sensitivity score for each patch.

