# OpenReview forum: "Can't See the Wood for the Trees: Can Visual Adversarial Patches Fool Hard-Label Large Vision-Language Models?"
_ICLR.cc/2025/Conference — ICLR 2025 Conference Withdrawn Submission_

### Official Review · Reviewer_KqjY · 2024-10-25

**Soundness:** 3
**Presentation:** 3
**Contribution:** 2
**Rating:** 5
**Confidence:** 4

**Summary:**

This paper addresses a very important issue in the field of multimodal security. Executing untargeted attacks on Visual Language Models is of greater practical significance compared to the widely studied white-box attacks in academia, as attackers often lack access to model parameters and architectural information. The proposed HardPatch method identifies the most critical visual tokens and then employs a query-based approach to execute adversarial attacks, making the entire pipeline sound very reasonable. Extensive evaluations on LVLM models and datasets are conducted to demonstrate the adversarial nature of the proposed HardPatch.

**Strengths:**

1. The writing of this paper is very smooth and easy to read.
2. The research question is highly relevant, as it is often difficult to obtain model weights and architectural information for the victim model in real-world scenarios. This study expands the understanding of the effectiveness of adversarial attacks in real-life contexts.
3. The experiments are comprehensive, with both comparative and ablation studies demonstrating the effectiveness of the proposed method.

**Weaknesses:**

line 131, simply using `\cite` will suffice. In line 310, "We" should be changed to "we."

One of my significant concerns is that, based on the cases presented in the paper, such as Figures 1 and 5, the adversarial images have undergone significant changes compared to the originals, making it easy for ordinary users to notice the anomalies. For example, these adversarial patches resemble embedded QR codes, which would raise suspicion among users. The standard definition of adversarial samples generally states that they involve adding perturbations that are imperceptible to the human eye. However, the noise generated by HardPatch clearly contradicts this assumption. Further clarification is needed on how to ensure that the noise remains imperceptible to the human eye.

**Questions:**

I would like to see further clarification in the Weaknesses. If the explanations are reasonable, I would be happy to revise my score.

---

### Official Review · Reviewer_uAdq · 2024-10-31

**Soundness:** 2
**Presentation:** 3
**Contribution:** 2
**Rating:** 5
**Confidence:** 5

**Summary:**

Motivated by the research gap and counter-practical phenomenon in previous methods, this paper proposes the first hard-label attack for LVLMs, named HardPatch, to generate visual adversarial patches by solely querying the model. Extensive empirical observations suggest that with appropriate patch substitution and optimization, HardPatch can craft effective adversarial images to attack hard-label LVLMs.

**Strengths:**

- This paper is well-written and easy to follow. The motivation of this paper is clear. The experiment results are good.
- This paper presents a respectable attempt to deceive LVLMs under more realistic hard-label settings.

**Weaknesses:**

- The attacks are only restricted to patch-based kinds. This will limit the scope and practicality of this method, given that adversarial patches are much easier to be detected by human eyes and consequently hinder the stealthiness, especially when multiple patches are applied to one image as the authors illustrated in line 190-191.
- The computational costs of this paper are still very considerable, with over 5 hours and 50 GB memory consumption for one complete attack. This will reduce the usefulness and versatility of this method.
- The reason of choosing the normalized uniform distribution $u \cdot exp(u-1)$ for $\Delta_k$ is unclear. It is not explained in this paper why the authors choose this function instead of the others. I hope that the authors could provide more empirical or theoretical analysis regarding this choice.

**Questions:**

- The ablation study of this paper is incomplete. In additional to the patch number and patch size that the authors have detailedly discussed and analyzed, how do the other hyper-parameters (i.e., optimization number $T$, additive noise number $K$, targeted attack similarity threshold $\tau_1$ and untargeted attack similarity threshold $\tau_2$) affect the attack results? How do the authors choose their values? Are there any heuristic insights about their impacts?
- I wonder how many queries are needed for one complete attack. If the number of queries is not tiny for one complete attack, could it be easily detected by the LVLMs and become more noticeable? Is it possible that the victims can effectively defend the proposed attack by simply checking the consecutive query counts of malicious users?
- The random noises $\delta_m$ and $\Delta_k$ seem to be a core part of the proposed method. What are the impacts of different randomization ways (e.g., varying the random seeds, changing the randomized functions) of $\delta_m$ and $\Delta_k$? Are the attack results sensitive to different randomization ways?

---

### Official Review · Reviewer_5TVg · 2024-10-31

**Soundness:** 3
**Presentation:** 3
**Contribution:** 2
**Rating:** 3
**Confidence:** 4

**Summary:**

Large Vison-Language Models (LVLMs) are susceptible to adversarial perturbations. However, recent adversarial attacks to LVLMs in ideal scenarios need gradient or rely on the additional knowledge of other surrogate LVLMs, which are not provided in real-world scenarios. This paper is the first adversarial patch attack being investigated in LVLM applications. It proposed a novel method called HardPatch, to generate visual adversarial patches by solely querying the model in hard-label setting. The method first split each image into uniform patches and mask each of them to individually assess their sensitivity to the LVLM model. Then it utilizes Monte Carlo method to obtain certain noise patch’s approximate gradient in statistics through numerous queries. The article conducted extensive experiments and obtained sufficient data to support its argument.

**Strengths:**

The article’s writing is fluent, the logic is very clear, and the inference is convincing. This paper provides a fairly in-depth discussion on the adversarial patch attack to LVLMs in strict black-box setting, and conducts comprehensive experiment. It not only designs adversarial patch attack to single image, but also extend the method into a universal attack setting. The experimental data is sufficient to support its argument. The innovation of the proposed method can be summarized as the following two points:
- Using black patche with 0 pixel value as mask to test the sensitivity of the victim model to patch at different positions in the image, which is based on the global semantic invariant characteristic with local contexts mask of MAE.
- In the phase of patch noise optimization, utilize Monte Carlo method to obtain noise patch’s approximate gradient optimization direction in statistics through a large number of queries, when the attackers are only allowed to access the output of LVLMs.

**Weaknesses:**

- Even if only attacking a single image, the number of queries is excessive, which greatly reduced the efficiency of the attack. And the high computational cost limits the scope of application of this method.
- The stealth of this attack method is poor. Excessive and frequent queries on LVLMs can easily be detected as abnormal on the server side, triggering the defense mechanism of large models. The masking area of the image is too large when multiple patches are applied.

**Questions:**

- The comparison with MF (Zhao et al.,2024) seems unfair, because the MF is adversarial perturbation and HardPatch is patch.
- HardPatch is likely to completely destroy the important semantic parts of the image when selecting the sensitive part and replacing it with patch.

---

### Official Review · Reviewer_ySFj · 2024-11-03

**Soundness:** 2
**Presentation:** 3
**Contribution:** 2
**Rating:** 3
**Confidence:** 4

**Summary:**

This paper introduces a novel adversarial attack for vision-language large models (VLLMs). The method follows a hard-label attack setting, meaning that the attacker only accesses the predicted text output of the target model. The approach first splits the input image into small patches and evaluates each patch's sensitivity, then prioritizes modifications to the most sensitive patches using zero-order optimization. The optimization objective is to maximize the similarity between the VLLM output and a target text. The paper evaluates the method on four open-source VLLMs in both untargeted and targeted attack settings. Empirical results demonstrate the effectiveness of the proposed method to some extent.

**Strengths:**

1. The proposed method is a black-box attack with the potential to target proprietary VLLMs like GPT-4o (although it has not yet yielded results in the current submission). Furthermore, this approach is well motivated and neat. It does not require many magic number hyper-parameters.

2. The empirical results show that the method can effectively attack all four VLLMs, even with a low noise size.

3. The paper is well written and easy to follow.

**Weaknesses:**

1. Results on recent SOTA open source VLLMs (Qwen2 VL, Llama3.2 Vision) and proprietary VLLMs (GPT4-o, Claude3.5) are lacking. Although the proposed method shows effectiveness on four victim VLLMs, these VLLMs (BLIP2, MiniGPT-4 and LLaVA-1.5) are small in size and less robust.

Another limitation of the victim VLLMs used is that they all use the same **frozen** image encoder (CLIP ViT), making it difficult to determine if the proposed method exploits this dependency (when computing the text similarity). Thus it is necessary to have a wider range of evaluation.

2. The paper claims that this is the first hard label attack. however, the transfer-based attack described in [1, 2] do not rely on any information about the target model, representing a more challenging scenario that can also be categorized as a hard label attack, since it has access to the text output but does not utilize it.

3. In the target attack setting, I believe that phrases like "I am sorry," "I do not know," or "I cannot answer" are not suitable targets. It is challenging to determine whether an attack was successful or merely an illusion stemming from inadequate VLLM model capabilities. In contrast, the target "Bomb" is appropriate, and I would like to see some visual example for targets like this, along with the model's response to the image, which is not included in the current submission. For example, what does the LLaVA generate for the captioning and VQA tasks when targeting "Bomb"? These results are necessary for people to judge whether an attack is successful or not.


[1] On evaluating adversarial robustness of large vision-language models. Advances in Neural Information Processing Systems, 36, 2024.

[2] How Robust is Google’s Bard to Adversarial Image Attacks? Workshop on Robustness of Few-shot and Zero-shot Learning in Foundation Models at NeurIPS 2023.

**Questions:**

1. Which $p$ norm are you using in Equation 1? It seems the method is limited to $\ell_1$ and $\ell_2$ attack, and cannot be applied to $\ell_\infty$ attack, since the method needs to completely change 1~4 patches. For example in Figure 5, I think the patch is completely changed, and the $\ell_\infty$ norm difference between the original image and the attacking image should be very large. Is it correct?


2. In Algorithm 1 and Equation 2-6, I did not see the perturbation budget $\varepsilon$. How do you keep the perturbation budget (or do not keep)?


3. In Table 2, when comparing with [1], I assume the paper compares with the result of ViT-B/32 from Table 2 in [1]. However, the attack in [1] is $\ell_\infty$ attack with $\varepsilon=8$ but you are using $\varepsilon=16$ with a unknown $\ell_p$ norm,  I do not think it is a fair comparison.

4. In Table 4 and 5, do you keep the perturbation budget  $\|x-x'\|_p\leq\varepsilon$. If yes, given the same perturbation budget, why having more perturbed pixels or more number of patches indicates an easier attack?

If that is the case, i.e., more pixels -> easier attack, a better experiments should be comparing:
 - 4 of $5\times5$ patches
- 8 of $7\times7$ patches: $8 \approx 4 / 5^2 / (1 / 7^2)$
- 13 of $9\times9$ patches: $13 \approx 4 / 5^2 / (1 / 9^2)$
in Table 5. Also adjust the patch size when comparing different number of patches.

5. In table 8, what does the GPU hours indicate. Is it GPU hours per image or the GPU hours for the entire dataset? If it is per image, I think this attack is hard to scale, at least less scalable to transferable attacks. If it is for the entire dataset, what is the GPU hour cost for a single image on a single H100 machine?


[1] On evaluating adversarial robustness of large vision-language models. Advances in Neural Information Processing Systems, 36, 2024.

---

### Official Review · Reviewer_TmPf · 2024-11-09

**Soundness:** 3
**Presentation:** 3
**Contribution:** 3
**Rating:** 6
**Confidence:** 4

**Summary:**

This paper introduces HardPatch, an adversarial attack method targeting large vision-language models (LVLMs) in a hard-label or black-box setting, where only the model’s input and output are accessible, with no access to model gradients or architecture details. HardPatch works by splitting an input image into patches and testing each patch for its sensitivity by measuring changes in LVLM output when patches are masked. Patches deemed most impactful are altered through a patch substitution and gradient estimation process using random noise to maximize the adversarial effect. This approach allows attackers to deceive LVLMs by carefully selecting and modifying only a few vulnerable patches instead of applying global noise, making it computationally efficient and less perceptible.

**Strengths:**

- The reviewer found the proposed method to be interesting. Specifically, the HardPatch approach strategically focuses on altering specific, vulnerable patches rather than adding noise to the entire image, making it computationally efficient and more practical for real-world applications. This localized approach also makes the attack harder to detect compared to traditional global noise methods.

- The reviewer appreciates the author's effort to evaluate HardPatch across several state-of-the-art LVLMs (e.g., BLIP-2, MiniGPT-4, LLaVA-1.5) and diverse datasets (e.g., MS-COCO, ImageNet), demonstrating the method's effectiveness across different models and image domains.

- Overall, the paper is well-written and easy to understand.

**Weaknesses:**

- The reviewer finds that HardPatch’s success is likely dependent on factors such as patch size, the number of patches, and their positions, yet the paper provides limited exploration of these parameters. This sensitivity could affect the method’s adaptability to different image types or model architectures.

- On a similar note, the approach of splitting images into multiple patches and testing each individually could become computationally intensive for higher-resolution images. The paper does not address how well HardPatch scales with large, complex images, potentially limiting its applicability in practical scenarios.

- While the authors discuss the effectiveness of HardPatch in bypassing defenses, they only briefly test against a simple rotation-based defense. A more comprehensive evaluation against various state-of-the-art defenses (e.g., adversarial training or advanced input transformations) could provide a stronger understanding of HardPatch’s robustness.

**Questions:**

N/A

---

### Note · Authors · 2024-11-13

I have read and agree with the venue's withdrawal policy on behalf of myself and my co-authors.